# Diastereoselective Synthesis of Dispiro[Imidazothiazolotriazine-Pyrrolidin-Oxindoles] and Their Isomerization Pathways in Basic Medium

**DOI:** 10.3390/ijms242216359

**Published:** 2023-11-15

**Authors:** Alexei N. Izmest′ev, Dmitry B. Vinogradov, Angelina N. Kravchenko, Natalya G. Kolotyrkina, Galina A. Gazieva

**Affiliations:** 1N. D. Zelinsky Institute of Organic Chemistry, Russian Academy of Sciences, 47 Leninsky Prosp., Moscow 119991, Russia; nebeli@mail.ru (A.N.I.); vinogradovdima14@ioc.ac.ru (D.B.V.); kani@ioc.ac.ru (A.N.K.); nkolotyr@gmail.com (N.G.K.); 2Department of General and Inorganic Chemistry, National University of Science and Technology “MISIS”, 4 Leninsky Prosp., Moscow 119049, Russia

**Keywords:** dispirooxindoles, [3+2] cycloaddition, imidazothiazolotriazines, azomethine ylides, epimerization, isomerization

## Abstract

Highly diastereoselective methods for the synthesis of two series of regioisomeric polynuclear dispyroheterocyclic compounds with five or six chiral centers, comprising moieties of pyrrolidinyloxindole and imidazo[4,5-*e*]thiazolo[3,2-*b*]-1,2,4-triazine of linear structure or imidazo[4,5-*e*]thiazolo[2,3-*c*]-1,2,4-triazine of angular structure, have been developed on the basis of a [3+2] cycloaddition of azomethine ylides to functionalized imidazothiazolotriazines. Depending on the structure of the ethylenic component, cycloaddition proceeds as an *anti-exo* process for linear derivatives, while cycloaddition to angular ones resulted in a *syn-endo* diastereomer. Novel pathways of isomerization for the synthesized *anti-exo* products upon treatment with sodium alkoxides have been found, which resulted in two more series of diastereomeric dispiro[imidazothiazolotriazine-pyrrolidin-oxindoles] inaccessible with the direct cycloaddition reaction. For the first series, the inversion of the configuration of one stereocenter, i.e., C-4′ atom of the pyrrolidine cycle, (epimerization) was established. For the second one, configuration of the obtained diastereomer formally corresponded to the *syn-endo* approach of the azomethine ylide in the case of cycloaddition to the ethylenic component.

## 1. Introduction

Recent trends of organic and medicinal chemistry consist in constructing rigidly oriented spiroheterocyclic structures with high solubility and bioavailability as well as the ability to interact effectively with various biological targets [1]. Special attention is paid to the oxindoles spiro-linked with the pyrrolidine cycle, which have become popular since the discovery of valuable pharmacological properties of a number of natural alkaloids, such as spirotriprostatin B [2], horsfilin [3] and mitraphyllin [4] at the end of the XX century. The antitumor activity of synthetic spiropyrrolidineoxindoles is actively investigated [5,6,7,8,9,10,11] (Figure 1). For example, polymerization inhibitors of actin [8] and tubulin [9], as well as MDM2–p53 protein–protein interaction [10,11,12,13,14] were obtained.

A convenient and effective method for the synthesis of spirooxindoles is the [3+2] cycloaddition reaction of azomethine ylides to unsaturated compounds [14,15,16,17,18,19,20,21,22,23,24,25,26,27]. Such reactions often proceed with high diastereoselectivity and allow to obtain products with a certain relative configuration, while other isomers remain unavailable.

At the same time, spiropyrrolidineoxindole **1** prepared using cycloaddition appeared to be a less active MDM2–p53 protein–protein interaction inhibitor than the **MI-888** prepared via base-induced isomerization of compound **1** (Figure 1A) [12]. Therefore, development of the methods for the isomerization of spiropyrrolidineoxindole prepared via a cycloaddition reaction are relevant.

Earlier, we have discovered the skeletal rearrangement of dispiropyrrolidineoxindoles **2** into isomers **3** upon treatment with KOH (see Figure 1B) [28]. Herein, we carried out the cycloaddition of ylidene derivatives of imidazothiazolotriazine and azomethine ylides generated from amino acids and isatins and studied various isomerization pathways of synthesized cycloaddition products **4** in basic medium (see Figure 1C).

## 2. Results and Discussion

Target dispiro[imidazothiazolotriazine-6,3′-pyrrolidin-2′,3″-oxindoles] **4a**–**k** were prepared according to earlier elaborated procedure [28,29] via a three-component reaction of [3+2] cycloaddition of azomethine ylides generated in situ from N-alkylamino acids **7a**–**d** and isatins **8a**–**c** with functionalized imidazothiazolotriazines **9a,b** [30] in boiling acetonitrile for 8 h (24 h for **4e,j**) (Figure 2). For completely conversing the initial ethylenic components **9a**,**b** and increasing the yields of the target dispyrocyclic compounds **4a**–**k**, a 1.5-fold excess of isatins and amino acids was used, since the particles formed from them have low stability and a short lifespan.

The highest yields of cycloadducts **4a**, **4b**, **4f** and **4g** (91, 84, 93 and 95%, respectively) were observed when using N-methyl- and N-ethylglycines **7a**,**b** as amino acid and unsubstituted isatin **8a**, while application of more sterically hindered N-isopropylglycine **7c** and substituted isatins **8b**,**c** led to some decrease in the yields of corresponding dispyrocyclic structures **4c**–**e** and **4h**–**j** to 74–82%. The introduction of the N-methyl norvaline derivative **7d** into the reaction, which has an additional substituent at the *α*-carbon atom in comparison with sarcosine, was accompanied by a significant decrease in the yield of the corresponding product **4k** to 31%. The relative configuration of compound *rel*-(2′*R*,3a*S*,4′*S*,6*R*,9a*R*)-**4f** was unambiguously assigned via single crystal X-ray diffraction and corresponded to the configuration of previously obtained similar compounds [28,29]. The configuration of all other products was assigned by analogy. The configuration of the pyrrolidine cycle C-5′ atom in the structure **4k** is adopted by analogy with the examples known in the literature [31,32]. The absence of signals from other isomers in the ^1^H NMR spectra of the evaporated reaction mixtures indicates a high selectivity of the reaction and the formation of single regioisomer and diastereomer **4**.

At the same time, the formation of the pyrrolidine cycle during the process of [3+2] cycloaddition of azomethine ylides to the double bond of ethylenic components is accompanied with the generation of three new stereocenters. Together with the stereocenters available in the initial compounds (C-3a and C-9a), the number of chiral centers can theoretically determine the formation of 16 diastereomers (16 enantiomeric pairs). The reasons for decrease in the number of possible diastereomers can be the following: (i) the use of compounds with a *Z*-configuration of double bonds and rigid *cis*-junction at the C-3a–C-9a bond as ethylenic components; (ii) the synchronicity of the cyclocondensation process; and (iii) nonequivalence in modes of approach of the azomethine ylide to ethylenic component.

Azomethine ylide generated in situ via condensation of isatin and amino acid (for example, N-methylglycine) followed by thermic decarboxylation of the intermediate lactone (Figure 3).

Meanwhile, the nitrogen containing three atom components involved in the [3+2] cycloaddition reaction can be attributed to the dipolar, zwitterionic, pseudo(mono)- or pseudodiradical type [33,34,35,36,37,38]. However, recent experimental and theoretical data obtained for azomethine ylides indicate their pseudoradical nature [33,34,35].

Theoretically possible mechanisms of the cycloaddition reaction could include both one-step and stepwise pathways for the formation of the pyrrolidine ring (Figure 4). During the stepwise reaction of azomethine ylides, which have both a dipole (Path I) and a pseudodiradical (Path II) character, free rotation around the bond brought from the ethylenic component to the target pyrrolidine ring is unlocked within the resulting intermediates. Therefore, a stepwise process should lead to the limitation of stereoselectivity and the formation of two stereoisomeric products **4a** and **5a**. The absence of signals from other isomers (including epimeric structures **5**) in the ^1^H NMR spectra of the evaporated reaction mixtures evidence of one-step process (for example, Path III or IV). Recent theoretical investigations (using the topological analysis of the electron localization function (ELF) at the B3LYP/6-31G(d) level of theory) of the cycloaddition reaction of symmetric azomethine ylide prove synchronous concerted transition state structure, and the process may be electronically classified as pseudo-diradical [2n + 2π] process (Path IV) [34]. However, the presence of electron-withdrawing carbonyl C=O group in the isatin derivative can modify its reactivity. By taking into account the presence of conjugated C=O group in an ethilenic component, it can be assumed that the cycloaddition take place through a polar non-concerted two-stage one-step mechanism associated with the nucleophilic attack of the least substituted carbon of azomethine ylide on the β-conjugated position of the ethylenic component **9** [27,35,36,37].

Additionally, the complicated structures, both three atom component and ethylenic component, propose nonequivalence in modes of approach of the azomethine ylide (Figure 5). The addition of sterically bulky azomethine ylides occurs from the less sterically loaded *anti*-side relative to the imidazolidine cycle and proceeds via an *exo*-transition state, where the carbonyl groups of the oxindole fragment and the thiazolidinone ring become relative to the pyrrolidine cycle on different sides.

One example shows that the introduction of chiral (*R*)-2-[(1-phenylethyl)amino]acetic acid **7e** into the reaction leads to the formation of a mixture of two diastereomers **4l** and **4m** in approximately equal amounts instead of racemate (Figure 6). With an increase in the bulk of the substituent in the reagent, the use of a two-fold excess of amino acid and isatin, as well as a longer reaction time (36 h) are required, and the total yield of the mixture of products **4l** and **4m** decreased to 41%. It is shown that diastereomers **4l** and **4m** have different retention times in the chromatographic column and can be isolated individually (see Appendix A).

It was noted above that the treatment of structurally related dispiro[imidazothiazolotriazine-pyrrolidin-oxindoles] **2**, having an aryl substituent in the pyrrolidine cycle, with KOH is accompanied by hydrolysis of the amide bond and skeletal rearrangement of the thiazolotriazine system, which resulted in regioisomeric products **3** [28]. In turn, boiling esters **4a**–**j** in alcohols in the presence of sodium alkoxides mainly led to the formation of a mixture of two new diastereomers **5** and **6** in different ratios (Figure 7). ^1^H NMR monitoring of the reaction showed that the complete conversion of the initial compounds **4** into products **5** and **6** was achieved with the action of 0.25 equivalents of sodium alkoxide for 4 h.

Herein, an increase in the reaction time had little effect on the yields and the ratio of the products formed, while an increase in the amount of base can accelerate the reaction and slightly change the ratio of products, reducing, nevertheless, their total yield. In some cases, compounds **5c**,**d**,**h** were isolated from the reaction mixture without impurities of other diastereomers. The signals of their isomers **6c**,**d**,**h** were observed in the ^1^H NMR spectra of evaporated reaction mixtures in trace amounts and were not isolated in these cases. Each of the isomers **5a**–**j** and **6a**,**b**,**e**–**g**,**i**,**j** was isolated individually via fractional crystallization from the reaction mixtures or MeCN. The relative configurations of the chiral centers of diastereomers *rel*-(2′*R*,3a*S*,4′*R*,6*R*,9a*R*)-**5** and *rel*-(2′*R*,3a*S*,4′*R*,6*S*,9a*R*)-**6** were unambiguously determined using single crystal X-ray diffraction for compounds **5b** and **6a**. For compounds **5**, the inversion of the configuration of one stereocenter, i.e., C-4′ atom of the pyrrolidine cycle (epimerization), was established in comparison with compounds **4**, which was previously described for related spiropyrrolidineindoles on a single example [39]. Configuration of diastereomers **6** indicated the inversion of two stereocenters compared to starting compounds **4** (C-3′(C-6) and C-4′ atoms of the pyrrolidine cycle) and formally corresponded to the *syn-endo* approach of the azomethine ylide in the case of cycloaddition to dipolarophile **9** (Figure 5).

We assumed that the presence of an electron-withdrawing ester group at 4′ position of compounds **4** makes hydrogen atom at the corresponding *α*-carbon atom acidic. Therefore, sodium alkoxide causes the primary deprotonation of structures **4** and the formation of carbanion **A** (Figure 8), which transforms into a more stable anion **B** due to the elimination of the thiolate anion. As a result of the opening of the thiazolidine cycle, it is possible to freely rotate the spiropyrrolidineoxindole fragment of the molecule around a single C–C bond, the further addition of the thiolate anion to a double bond of the Michael acceptor, and finally, the spiro node formation in new *syn-endo*-diastereomers **6** inaccessible via the direct cycloaddition reaction. The processes occurring in this case do not affect other asymmetric centers present in the molecule (C-3a, C-9a and C-2′); therefore, the corresponding carbon atoms in isomeric structures **4**, **5** and **6** have the same configuration. The *anti-exo* epimer **5** can be form from carbanion **A** and alcohol.

The structures of the prepared compounds were also proven using spectral methods. In the ^1^H NMR spectra, a characteristic signal that allows the resulting compound to be assigned to one of the diastereomeric products **4**, **5**, or **6** is the signal of the 4′-CH proton of the pyrrolidine ring, which experiences different deshielding effects from neighboring carbonyl groups. Due to its closer spatial arrangement to the atom 4′-CH, deshielding effect of the carbonyl group of the oxindole fragment is higher than that of the carbonyl group of the thiazolidinone ring. As a result, the corresponding signal for the epimeric products **5** downfield shifted (4.40–4.45 ppm) compared to its location in the spectra of starting structures **4** (4.03–4.10 ppm) (Figure 2). In *syn-endo* diastereomers **6**, the carbonyl groups of the oxindole fragment and the thiazolidine ring are on the same side relative to the pyrrolidine ring, which leads to maximum deshielding of the 4′-CH hydrogen atom by both groups and a strong downfield shift of its signal to the region of 5.02–5.07 ppm.

To obtain skeletal isomers of compounds **4**, three-component [3+2] cycloaddition reaction of azomethine ylides with functionalized imidazothiazolotriazines **10a**,**b [30]** of an angular structure was also carried out by boiling the starting compounds in acetonitrile. The previously unknown regioisomeric dispirocyclic structures **11a**–**f** were synthesized in 41–73% yields (Figure 9). The relative configuration of the structure *rel*-(2′*S*,3a*R*,4′*S*,7*R*,9a*S*)-**11e** was determined via X-ray diffraction analysis and appeared to be corresponding to a *syn-endo* diastereomer. Chemical shifts (4.72–4.82 ppm) of the signal for the 4′-CH hydrogen atom of the pyrrolidine ring in the ^1^H NMR spectra of compounds **11a**–**f** allow to assign all compounds to the diastereomers of the same structure.

## 3. Materials and Methods

### 3.1. General Information

All standard reagents were purchased from Aldrich or Acros Organics and used without further purification.

Melting points were determined on a Stuart SMP20 apparatus (Stuart (Bibby Scientific), Stone, UK). 

IR spectra were recorded on a Bruker “Alpha” spectrophotometer (Bruker, Billerica, MA, USA) in the range 400–4000 cm^−1^ (resolution: 2 cm^−1^).

^1^H and ^13^C NMR spectra were recorded on a Bruker AM-300 (Bruker Biospin, Ettlingen, Germany; 300.13 and 75.47 MHz, respectively) and Bruker AV600 (Bremen, Germany; 150.90 MHz (^13^C)) spectrometers and referenced to the residual solvent peak. The chemical shifts are reported in ppm (δ); multiplicities are indicated by s (singlet), d (doublet), t (triplet), q (quartet), m (multiplet) and dd (doublet of doublets). Coupling constants, *J*, are reported in Hertz. 

High-resolution mass spectra (HRMS) were measured on the Bruker micrOTOF II instrument (Bruker Daltonik GmbH, Bremen, Germany) using electrospray ionization (ESI) or on Agilent 7890A GC coupled with Waters GCT Premier orthogonal acceleration time-of-flight detector using electron impact (EI) ionization. The measurements were conducted in a positive ion mode (interface capillary voltage: 4500 V); mass range from *m*/*z* 50 to 3000 Da; external or internal calibration was done with Electrospray Calibrant Solution (Fluka Analytical/Sigma Aldrich, Steinheim, Germany). A syringe injection was used for solutions in MeCN or MeOH (flow rate 3 μL/min). N_2_ was applied as a dry gas; interface temperature was set at 180 °C.

HPLC-MS analysis was performed on an Agilent 1200 instrument (Agilent Technologies, Santa Clara, CA, USA) under the following conditions: Reprosil-Pur Basic C18 (250 × 4.6 mm) column, 5 μm (Dr. Maisch GmbH); eluents: 0.01% CF_3_COOH–H_2_O (A), 0.01% CF_3_COOH–MeCN (B); 1 mL/min flow rate. The analyzed compounds were detected using a spectrophotometric detector at λ = 220 nm. Preparative HPLC separation of reaction products was carried out on a semipreparative system with Gilson pumps (blocks 305 and 306), Gilson 805 manometric module, Jetchrom UVV-105 detector, ReproSil Pure Basic C 18 column (250 × 20 mm), 10 nm; 10 mL/min flow rate, with UV detection at λ 220 nm.

The starting dipolarophiles **9a**,**b** and **10a**,**b** were prepared according to a procedure described in the literature [30]. Amino acid **7e** was prepared according to a procedure mentioned in the literature [40].

### 3.2. General Procedure for the Synthesis of Compounds ***4a**–**m***

A mixture of corresponding compound **9a**,**b** (1 mmol), aminoacetic acid **7a***–***d** (1.5 mmol) and isatin **8a***–***c** (1.5 mmol) in MeCN (20 mL) was refluxed with stirring for 8 h (24 h for **4e**,**j**). After cooling, the precipitate of compounds **4a***–***k** was filtered off, washed with methanol and dried at 50 °C.

For the synthesis of diastereomers **4l** and **4m,** the mixture of compound **9b** (1 mmol), isatin **8a** and (*R*)-2-[(1-phenylethyl)amino]acetic acid **7e** in MeCN (40 mL) reflux for 12 h, an additional portion of isatin (1 mmol) and amino acid (1 mmol) was added and was continued to reflux for 24 h. After cooling, the precipitate of mixture of compounds **4l** and **4m** was filtered off, washed with methanol, and dried at 50 °C.

**Methyl *rac*-(2′*R*,3a*S*,4′*S*,6*R*,9a*R*)-1,1′,3-trimethyl-2,2″,7-trioxo-3a,9a-diphenyl-1,2,3,3a,9,9a-hexahydro-7*H*-dispiro[imidazo[4,5-*e*]thiazolo[3,2-*b*][1,2,4]triazine-6,3′-pyrrolidine-2′,3″-indoline]-4′-carboxylate (4a).** Yield 579 mg (91%); white powder; mp: 231–232 °C. IR (KBr): ν 3242, 3175 (NH), 3094, 3034 (Ar), 2950, 2871, 2838 (Alk), 1755, 1729, 1690, 1645, 1621, 1586 (C=O, C=N) cm^−1^. ^1^H NMR (300 MHz, DMSO-*d*_6_): δ 2.04 (s, 3H, 1′-NCH_3_), 2.52 (s, 3H, NCH_3_), 2.64 (s, 3H, NCH_3_), 3.38 (t, *J* = 8.6 Hz, 1H, 5′-H), 3.69 (s, 3H, OCH_3_), 3.84 (t, *J* = 8.9 Hz, 1H, 5′-H), 4.10 (t, *J* = 8.5 Hz, 1H, 4′-H), 6.39 (d, *J* = 7.2 Hz, 2H, Ph-2,6), 6.67 (d, *J* = 7.2 Hz, 2H, Ph-2,6), 6.89–7.20 (m, 8H, 2Ph-3-5, 5″-H, 7″-H), 7.30–7.50 (m, 2H, 4″-H, 6″-H), 7.64 (s, 1H, 9-NH), 10.77 (s, 1H, 1″-NH). ^13^C NMR (75 MHz, DMSO-*d*_6_): δ 25.00, 25.84 (2NCH_3_), 34.78 (1′-NCH_3_), 50.80, 51.97, 52.45 (C-4′, C-5′, OCH_3_), 63.65 (C-6), 78.61, 79.67, 82.18 (C-2′, C-3a, C-9a), 109.87 (C-7″), 122.17, 122.92, 126.51, 127.13, 127.44, 127.65, 128.07, 130.51 (2Ph-2-6, C-3a″, C-4″, C-5″, C-6″), 133.90, 134.68 (2Ph-1), 143.87, 147.87 (C-7a″, 4a-C=N), 159.10 (2-C=O), 166.37 (7-C=O), 169.06 (COOMe), 175.36 (2″-C=O). HRMS (ESI): *m*/*z* [*M* + H]^+^ calcd for C_33_H_31_N_7_O_5_S: 638.2180; found: 638.2189.

**Methyl *rac*-(2′*R*,3a*S*,4′*S*,6*R*,9a*R*)-1′-ethyl-1,3-dimethyl-2,2″,7-trioxo-3a,9a-diphenyl-1,2,3,3a,9,9a-hexahydro-7*H*-dispiro[imidazo[4,5-*e*]thiazolo[3,2-*b*][1,2,4]triazine-6,3′-pyrrolidine-2′,3″-indoline]-4′-carboxylate (4b).** Yield 546 mg (84%); white powder; mp: 247–249 °C. IR (KBr): ν 3250 (NH), 3186, 3094, 3062, 3032 (Ar), 2948, 2866, 2815 (Alk), 1730, 1693, 1637, 1586 (C=O, C=N) cm^−1^. ^1^H NMR (300 MHz, DMSO-*d*_6_): δ 0.93 (t, *J* = 7.2 Hz, 3H, CH_3_), 2.00–2.18 (m, 1H, 1′-NCH_2_), 2.20–2.31 (m, 1H, 1′-NCH_2_), 2.50 (s, 3H, NCH_3_), 2.63 (s, 3H, NCH_3_), 3.51 (t, *J* = 8.6 Hz, 1H, 5′-H), 3.63–3.78 (m, 4H, 5′-H, OCH_3_), 4.08 (t, *J* = 8.6 Hz, 1H, 4′-H), 6.36 (d, *J* = 7.1 Hz, 2H, Ph-2,6), 6.66 (d, *J* = 7.1 Hz, 2H, Ph-2,6), 6.90–7.20 (m, 8H, 2Ph-3-5, 5″-H, 7″-H), 7.35–7.50 (m, 2H, 4″-H, 6″-H), 7.63 (s, 1H, 9-NH), 10.75 (s, 1H, 1″-NH). ^13^C NMR (75 MHz, DMSO-*d*_6_): δ 14.11 (CH_3_), 25.53, 26.37 (2NCH_3_), 43.22 (1′-NCH_2_), 50.06, 51.08, 52.54 (C-4′, C-5′, OCH_3_), 64.07 (C-6), 79.05, 80.12, 82.63 (C-2′, C-3a, C-9a), 110.36 (C-7″), 122.71, 123.98, 127.02, 127.68, 127.98, 128.20, 128.64, 130.98 (2Ph-2-6, C-3a″, C-4″, C-5″, C-6″), 134.37, 135.17 (2Ph-1), 144.22, 148.52 (C-7a″, 4a-C=N), 159.66 (2-C=O), 166.86 (7-C=O), 169.73 (COOMe), 176.21 (2″-C=O). HRMS (ESI): *m*/*z* [*M* + H]^+^ calcd for C_34_H_33_N_7_O_5_S: 652.2337; found: 652.2334.

**Methyl *rac*-(2′*R*,3a*S*,4′*S*,6*R*,9a*R*)-1′-isopropyl-1,3-dimethyl-2,2″,7-trioxo-3a,9a-diphenyl-1,2,3,3a,9,9a-hexahydro-7*H*-dispiro[imidazo[4,5-*e*]thiazolo[3,2-*b*][1,2,4]triazine-6,3′-pyrrolidine-2′,3″-indoline]-4′-carboxylate (4c).** Yield 512 mg (77%); white powder; mp: 261–263 °C. IR (KBr): ν 3252, 3188 (NH), 3060, 3031 (Ar), 2954, 2878 (Alk), 1731, 1691, 1641, 1586 (C=O, C=N) cm^−1^. ^1^H NMR (300 MHz, DMSO-*d*_6_): δ 0.88 (d, *J* = 7.0 Hz, 3H, CH_3_), 0.93 (d, *J* = 6.9 Hz, 3H, CH_3_), 2.50 (s, 3H, NCH_3_), 2.59–2.72 (m, 4H, NCH_3_, 1′-NCH), 3.45 (t, *J* = 8.2 Hz, 1H, 5′-H), 3.68 (s, 3H, OCH_3_), 3.93 (t, *J* = 8.1 Hz, 1H, 5′-H), 4.04 (t, *J* = 8.0 Hz, 1H, 4′-H), 6.33 (d, *J* = 8.2 Hz, 2H, Ph-2,6), 6.65 (d, *J* = 7.9 Hz, 2H, Ph-2,6), 6.92–7.18 (m, 8H, 2Ph-3-5, 5″-H, 7″-H), 7.35 (d, *J* = 7.9 Hz, 1H, 4″-H), 7.43 (t, *J* = 7.7 Hz, 1H, 6″-H), 7.62 (s, 1H, 9-NH), 10.73 (s, 1H, 1″-NH). ^13^C NMR (75 MHz, DMSO-*d*_6_): δ 17.98, 22.62 (2CH_3_), 24.96, 25.82 (2NCH_3_), 44.49 (1′-NCH), 46.74, 49.39, 51.95 (C-4′, C-5′, OCH_3_), 63.84 (C-6), 77.19, 79.60, 82.32 (C-2′, C-3a, C-9a), 109.99 (C-7″), 121.96, 123.84, 126.58, 127.10, 127.40, 127.60, 128.05, 130.33 (2Ph-2-6, C-3a″, C-4″, C-5″, C-6″), 133.89, 134.76 (2Ph-1), 143.57, 147.85 (C-7a″, 4a-C=N), 159.11 (2-C=O), 165.91 (7-C=O), 169.45 (COOMe), 177.82 (2″-C=O). HRMS (ESI): *m*/*z* [*M* + H]^+^ calcd for C_35_H_35_N_7_O_5_S: 666.2493; found 666.2482.

**Methyl *rac*-(2′*R*,3a*S*,4′*S*,6*R*,9a*R*)-5″-bromo-1′-ethyl-1,3-dimethyl-2,2″,7-trioxo-3a,9a-diphenyl-1,2,3,3a,9,9a-hexahydro-7*H*-dispiro[imidazo[4,5-*e*]thiazolo[3,2-*b*][1,2,4]triazine-6,3′-pyrrolidine-2′,3″-indoline]-4′-carboxylate (4d).** Yield 570 mg (78%); white powder; mp: 260–262 °C. IR (KBr): ν 3252 (NH), 3059, 3034 (Ar), 2981, 2943, 2880, 2840 (Alk), 1733, 1690, 1640, 1585 (C=O, C=N) cm^−1^. ^1^H NMR (300 MHz, DMSO-*d*_6_): δ 0.95 (t, *J* = 7.1 Hz, 3H, CH_3_), 2.07–2.18 (m, 1H, 1′-NCH_2_), 2.23–2.34 (m, 1H, 1′-NCH_2_), 2.51 (s, 3H, NCH_3_), 2.66 (s, 3H, NCH_3_), 3.57 (t, *J* = 8.7 Hz, 1H, 5′-H), 3.63–3.68 (m, 4H, 5′-H, OCH_3_), 4.07 (t, *J* = 8.4 Hz, 1H, 4′-H), 6.31 (d, *J* = 7.3 Hz, 2H, Ph-2,6), 6.73 (d, *J* = 7.5 Hz, 2H, Ph-2,6), 6.91–7.16 (m, 7H, 2Ph-3-5, 7″-H), 7.43 (d, *J* = 2.0 Hz, 1H, 4″-H), 7.66 (dd, *J* = 8.3, 2.1 Hz, 1H, 6″-H), 7.89 (s, 1H, 9-NH), 10.95 (s, 1H 1″-NH). ^13^C NMR (75 MHz, DMSO-*d*_6_): δ 13.67 (CH_3_), 25.14, 25.93 (2NCH_3_), 42.97, 49.41, 50.11, 52.13 (1′-NCH_2_, C-4′, C-5′, OCH_3_), 63.26 (C-6), 78.77, 79.64, 82.49 (C-2′, C-3a, C-9a), 111.99, 114.17 (5″-CBr, C-7″), 125.79, 126.81, 127.20, 127.36, 127.66, 127.70, 128.11, 129.90 (2-Ph-2-6, C-4″, C-5″), 133.42, 134.00, 134.78 (2Ph-1, C-3a″), 143.35, 147.67 (7a″-C, 4a-C=N), 159.16 (2-C=O), 165.89 (7-C=O), 169.31 (COOMe), 175.40 (2″-C=O). HRMS (ESI): *m*/*z* [*M* + H]^+^ calcd for C_34_H_32_BrN_7_O_5_S: 730.1442; found 730.1448.

**Methyl *rac*-(2′*R*,3a*S*,4′*S*,6*R*,9a*R*)-6″-chloro-1′-ethyl-1,3-dimethyl-2,2″,7-trioxo-3a,9a-diphenyl-1,2,3,3a,9,9a-hexahydro-7*H*-dispiro[imidazo[4,5-*e*]thiazolo[3,2-*b*][1,2,4]triazine-6,3′-pyrrolidine-2′,3″-indoline]-4′-carboxylate (4e).** Yield 527 mg (77%); white powder; mp: 260–261 °C. IR (KBr): ν 3296, 3216 (NH), 3066, 3032 (Ar), 2977, 2936, 2874, 2817 (Alk), 1745, 1726, 1686, 1641, 1616 (C=O, C=N) cm^−1^. ^1^H NMR (300 MHz, DMSO-*d*_6_): δ 0.94 (t, *J* = 7.2 Hz, 3H, CH_3_), 2.00–2.19 (m, 1H, 1′-NCH_2_), 2.20–2.40 (m, 1H, 1′-NCH_2_), 2.51 (s, 3H, NCH_3_), 2.62 (s, 3H, NCH_3_), 3.49 (t, *J* = 8.5 Hz, 1H, 5′-H), 3.68 (m, 4H, 5′-H, OCH_3_), 4.08 (t, *J* = 8.5 Hz, 1H, 4′-H), 6.47 (d, *J* = 7.1 Hz, 2H, Ph-2,6), 6.68 (d, *J* = 7.1 Hz, 2H, Ph-2,6), 6.95 (d, *J* = 2.0 Hz, 1H, 7″-H), 6.98–7.20 (m, 7H, 2Ph-3-5, 5″-H), 7.39 (d, *J* = 8.3 Hz, 1H, 4″-H), 7.53 (s, 1H, 9-NH), 10.94 (s, 1H 1″-NH). ^13^C NMR (75 MHz, DMSO-*d*_6_): δ 13.58 (CH_3_), 24.98, 25.83 (2NCH_3_), 42.68 (1′-NCH_2_), 49.41, 51.07, 52.03 (C-4′, C-5′, OCH_3_), 63.56 (C-6), 78.16, 79.77, 81.91 (C-2′, C-3a, C-9a), 109.97 (C-7″), 122.03, 122.57, 126.40, 127.13, 127.51, 127.72, 128.12, 129.43 (2Ph-2-6, C-3a″, C-4″, C-5″), 133.89, 134.68, 134.85 (2Ph-1, 6″-CCl), 145.32, 148.05 (C-7a″, 4a-C=N), 159.06 (2-C=O), 166.35 (7-C=O), 169.01 (COOMe), 175.70 (2″-C=O). HRMS (ESI): *m*/*z* [*M* + H]^+^ calcd for C_34_H_32_ClN_7_O_5_S: 686.1947; found 686.1939.

**Ethyl *rac*-(2′*R*,3a*S*,4′*S*,6*R*,9a*R*)-1,1′,3-trimethyl-2,2″,7-trioxo-3a,9a-diphenyl-1,2,3,3a,9,9a-hexahydro-7*H*-dispiro[imidazo[4,5-*e*]thiazolo[3,2-*b*][1,2,4]triazine-6,3′-pyrrolidine-2′,3″-indoline]-4′-carboxylate (4f).** Yield 606 mg (93%); white powder; mp: 222–225 °C. IR (KBr): ν 3368, 3237 (NH), 3060 (Ar), 2975, 2942, 2875, 2790 (Alk), 1711, 1643 (C=O, C=N) cm^−1^. ^1^H NMR (300 MHz, DMSO-*d*_6_): δ 1.20 (t, *J* = 7.1 Hz, 3H, CH_3_), 2.04 (s, 3H, 1′-NCH_3_), 2.50 (s, 3H, NCH_3_), 2.61 (s, 3H, NCH_3_), 3.38 (t, *J* = 8.8 Hz, 1H, 5′-H), 3.83 (t, *J* = 8.9 Hz, 1H, 5′-H), 3.99–4.31 (m, 3H, 4′-H, OCH_2_), 6.29 (d, *J* = 7.6 Hz, 2H, Ph-2,6), 6.62 (d, *J* = 7.6 Hz, 2H, Ph-2,6), 6.89–7.20 (m, 8H, 2Ph-3-5, 5″-H, 7″-H), 7.33 (d, *J* = 7.5 Hz, 1H, 4″-H), 7.45 (t, *J* = 7.6 Hz, 1H, 6″-H), 7.74 (s, 1H, 9-NH), 10.76 (s, 1H, 1″-NH). ^13^C NMR (75 MHz, DMSO-*d*_6_): δ 13.98 (CH_3_), 25.04, 25.81 (2NCH_3_), 34.83 (1′-NCH_3_), 49.71, 52.69 (C-4′, C-5′), 60.83 (OCH_2_), 63.55 (C-6), 78.92, 79.45, 82.54 (C-2′, C-3a, C-9a), 109.86 (C-7″), 122.11, 122.77, 126.66, 127.11, 127.29, 127.35, 127.57, 128.06, 130.52 (2Ph-2-6, C-3a″, C-4″, C-5″, C-6″), 133.87, 134.64 (2Ph-1), 143.93, 147.56 (C-7a″, 4a-C=N), 159.03 (2-C=O), 166.18 (7-C=O), 168.58 (COOEt), 175.44 (2″-C=O). HRMS (ESI): *m*/*z* [*M* + H]^+^ calcd for C_34_H_33_N_7_O_5_S: 652.2337; found: 652.2327.

**Ethyl *rac*-(2′*R*,3a*S*,4′*S*,6*R*,9a*R*)-1′-ethyl-1,3-dimethyl-2,2″,7-trioxo-3a,9a-diphenyl-1,2,3,3a,9,9a-hexahydro-7*H*-dispiro[imidazo[4,5-*e*]thiazolo[3,2-*b*][1,2,4]triazine-6,3′-pyrrolidine-2′,3″-indoline]-4′-carboxylate (4g).** Yield 631 mg (95%); white powder; mp: 211–213 °C. IR (KBr): ν 3359, 3162 (NH), 3060 (Ar), 2970, 2940, 2886 (Alk), 1725, 1641 (C=O, C=N) cm^−1^. ^1^H NMR (300 MHz, DMSO-*d*_6_): δ 0.93 (t, *J* = 7.1 Hz, 3H, CH_3_), 1.20 (t, *J* = 7.0 Hz, 3H, CH_3_), 2.00–2.38 (m, 2H, 1′-NCH_2_), 2.50 (s, 3H, NCH_3_), 2.61 (s, 3H, NCH_3_), 3.52 (t, *J* = 8.3 Hz, 1H, 5′-H), 3.73 (t, *J* = 8.7 Hz, 1H, 5′-H), 4.05 (t, *J* = 8.5 Hz, 1H, 4′-H), 4.10–4.30 (m, 2H, OCH_2_), 6.29 (d, *J* = 7.6 Hz, 2H, Ph-2,6), 6.62 (d, *J* = 7.4 Hz, 2H, Ph-2,6), 6.90–7.16 (m, 8H, 2Ph-3-5, 5″-H, 7″-H), 7.34 (d, *J* = 7.5 Hz, 1H, 4″-H), 7.44 (t, *J* = 7.4 Hz, 1H, 6″-H), 7.71 (s, 1H, 9-NH), 10.76 (s, 1H, 1″-NH). ^13^C NMR (75 MHz, DMSO-*d*_6_): δ 13.64, 14.02 (2CH_3_), 25.06, 25.84 (2NCH_3_), 42.80 (1′-NCH_2_), 49.51, 49.82 (C-4′, C-5′), 60.87 (OCH_2_), 63.45 (C-6), 78.82, 79.46, 82.53 (C-2′, C-3a, C-9a), 109.87 (C-7″), 122.10, 123.39, 126.68, 127.16, 127.30, 127.39, 127.63, 128.11, 130.47 (2Ph-2-6, C-3a″, C-4″, C-5″, C-6″), 133.89, 134.67 (2Ph-1), 143.96, 147.66 (C-7a″, 4a-C=N), 159.06 (2-C=O), 166.11 (7-C=O), 168.74 (COOEt), 175.85 (2″-C=O). HRMS (ESI): *m*/*z* [*M* + H]^+^ calcd for C_35_H_35_N_7_O_5_S: 666.2493; found: 666.2484.

**Ethyl *rac*-(2′*R*,3a*S*,4′*S*,6*R*,9a*R*)-1′-isopropyl-1,3-dimethyl-2,2″,7-trioxo-3a,9a-diphenyl-1,2,3,3a,9,9a-hexahydro-7*H*-dispiro[imidazo[4,5-*e*]thiazolo[3,2-*b*][1,2,4]triazine-6,3′-pyrrolidine-2′,3″-indoline]-4′-carboxylate (4h).** Yield 544 mg (80%); white powder; mp: 209–210 °C. IR (KBr): ν 3241, 3186 (NH), 3069, 3033 (Ar), 2967, 2933, 2897, 2833 (Alk), 1717, 1644 (C=O, C=N) cm^−1^. ^1^H NMR (300 MHz, DMSO-*d*_6_): δ 0.90 (d, *J* = 6.7 Hz, 3H, CH_3_), 0.96 (d, *J* = 6.5 Hz, 3H, CH_3_), 1.23 (t, *J* = 7.1 Hz, 3H, CH_3_), 2.51 (s, 3H, NCH_3_), 2.54–2.71 (m, 4H, NCH_3_, 1′-NCH), 3.48 (t, *J* = 7.6 Hz, 1H, 5′-H), 3.93–4.04 (m, 2H, 4′-H, 5′-H), 4.15–4.23 (m, 2H, OCH_2_), 6.28 (d, *J* = 7.5 Hz, 2H, Ph-2,6), 6.63 (d, *J* = 7.6 Hz, 2H, Ph-2,6), 6.90–7.21 (m, 8H, 2Ph-3-5, 5″-H, 7″-H), 7.34 (d, *J* = 7.5 Hz, 1H, 4″-H), 7.47 (t, *J* = 7.7 Hz, 1H, 6″-H), 7.70 (s, 1H, 9-NH), 10.76 (s, 1H, 1″-NH). ^13^C NMR (150 MHz, DMSO-*d*_6_): δ 14.09, 18.12, 22.78 (3CH_3_), 25.11, 25.90 (2NCH_3_), 44.79 (1′-NCH), 46.92, 48.61 (C-4′, C-5′), 60.89 (OCH_2_), 63.81 (C-6), 77.49, 79.47, 82.66 (C-2′, C-3a, C-9a), 110.07 (C-7″), 122.03, 123.82, 126.78, 127.17, 127.20, 127.42, 127.66, 127.71, 128.16, 130.44 (2Ph-2-6, C-3a″, C-4″, C-5″, C-6″), 133.94, 134.80 (2Ph-1), 143.67, 147.74 (C-7a″, 4a-C=N), 159.15 (2-C=O), 165.87 (7-C=O), 169.07 (COOEt), 178.01 (2″-C=O). HRMS (ESI): *m*/*z* [*M* + H]^+^ calcd for C_36_H_37_N_7_O_5_S: 680.2650; found: 680.2632.

**Ethyl *rac*-(2′*R*,3a*S*,4′*S*,6*R*,9a*R*)-5″-bromo-1′-ethyl-1,3-dimethyl-2,2″,7-trioxo-3a,9a-diphenyl-1,2,3,3a,9,9a-hexahydro-7*H*-dispiro[imidazo[4,5-*e*]thiazolo[3,2-*b*][1,2,4]triazine-6,3′-pyrrolidine-2′,3″-indoline]-4′-carboxylate (4i).** Yield 610 mg (82%); white powder; mp: 244–245 °C. IR (KBr): ν 3247, 3188, 3152 (NH), 3061, 3034 (Ar), 2981, 2938, 2872, 2823 (Alk), 1731, 1689, 1638, 1584 (C=O, C=N) cm^−1^. ^1^H NMR (300 MHz, DMSO-*d*_6_): δ 0.97 (t, *J* = 7.1 Hz, 3H, CH_3_), 1.23 (t, *J* = 7.0 Hz, 3H, CH_3_), 2.08–2.20 (m, 1H, 1′-NCH_2_), 2.29–2.36 (m, 1H, 1′-NCH_2_), 2.52 (s, 3H, NCH_3_), 2.64 (s, 3H, NCH_3_), 3.58–3.71 (m, 2H, 5′-H), 4.06 (t, J = 8.3 Hz, 1H, 4′-H), 4.14–4.24 (m, 2H, OCH_2_), 6.27 (d, *J* = 7.4 Hz, 2H, Ph-2,6), 6.74 (d, *J* = 7.4 Hz, 2H, Ph-2,6), 6.90–7.20 (m, 7H, 2Ph-3-5, 7″-H), 7.42 (d, *J* = 2.1 Hz, 1H, 4″-H), 7.70 (d, *J* = 8.2 Hz, 1H, 6″-H), 8.00 (s, 1H, 9-NH), 10.93 (s, 1H, 1″-NH). ^13^C NMR (75 MHz, DMSO-*d*_6_): δ 13.60, 13.95 (2CH_3_), 25.09, 25.83 (2NCH_3_), 42.97 (1′-NCH_2_), 48.39, 50.30 (C-4′, C-5′), 60.88 (OCH_2_), 63.12 (C-6), 78.91, 79.38, 82.77 (C-2′, C-3a, C-9a), 111.89, 114.07 (5″-CBr, C-7″), 125.60, 126.88, 127.10, 127.20, 127.51, 127.60, 128.01, 129.49 (2Ph-2-6, C-3a″, C-4″), 133.35, 133.89, 134.67 (2Ph-1, C-6″), 143.29, 147.33 (C-7a″, 4a-C=N), 159.01 (2-C=O), 165.59 (7-C=O), 168.75 (COOEt), 175.35 (2″-C=O). HRMS (ESI): *m*/*z* [*M* + H]^+^ calcd for C_35_H_34_BrN_7_O_5_S: 744.1598; found: 744.1586.

**Ethyl *rac*-(2′*R*,3a*S*,4′*S*,6*R*,9a*R*)-6″-chloro-1′-ethyl-1,3-dimethyl-2,2″,7-trioxo-3a,9a-diphenyl-1,2,3,3a,9,9a-hexahydro-7*H*-dispiro[imidazo[4,5-*e*]thiazolo[3,2-*b*][1,2,4]triazine-6,3′-pyrrolidine-2′,3″-indoline]-4′-carboxylate (4j).** Yield 517 mg (74%); white powder; mp: 190–192 °C. IR (KBr): ν 3293, 3213 (NH), 3064, 3031 (Ar), 2979, 2934, 2856, 2815 (Alk), 1731, 1688, 1645, 1613 (C=O, C=N) cm^−1^. ^1^H NMR (300 MHz, DMSO-*d*_6_): δ 0.94 (t, *J* = 7.2 Hz, 3H, CH_3_), 1.21 (t, *J* = 7.0 Hz, 3H, CH_3_), 2.08–2.16 (m, 1H, 1′-NCH_2_), 2.25–2.32 (m, 1H, 1′-NCH_2_), 2.51 (s, 3H, NCH_3_), 2.62 (s, 3H, NCH_3_), 3.51 (t, *J* = 8.5 Hz, 1H, 5′-H), 3.70 (t, *J* = 8.8 Hz, 1H, 5′-H), 4.06 (t, *J* = 8.6 Hz, 1H, 4′-H), 4.11–4.31 (m, 2H, OCH_2_), 6.41 (d, *J* = 7.1 Hz, 2H, Ph-2,6), 6.66 (d, *J* = 7.1 Hz, 2H, Ph-2,6), 6.90–7.20 (m, 8H, 2Ph-3-5, 5**″**-H, 7**″**-H), 7.36 (d, *J* = 8.2 Hz, 1H, 4″-H), 7.64 (s, 1H, 9-NH), 10.94 (s, 1H, 1″-NH). ^13^C NMR (150 MHz, DMSO-*d*_6_): δ 13.70, 14.08 (2CH_3_), 25.12, 25.87 (2NCH_3_), 42.83 (1′-NCH_2_), 49.69, 50.18 (C-4′, C-5′), 61.03 (OCH_2_), 63.42 (C-6), 78.47, 79.64, 82.24 (C-2′, C-3a, C-9a), 110.04 (C-7″), 122.09, 122.53, 126.61, 127.22, 127.53, 127.77, 127.83, 128.22, 129.19 (2Ph-2-6, C-3a″, C-4″, C-5″), 133.91, 134.67, 134.97 (2Ph-1, 6″-CCl), 145.44, 147.96 (C-7a″, 4a-C=N), 159.04 (2-C=O), 166.28 (7-C=O), 168.61 (COOEt), 175.80 (2″-C=O). HRMS (ESI): *m*/*z* [*M* + H]^+^ calcd for C_35_H_34_ClN_7_O_5_S: 700.2103; found: 700.2110.

**Ethyl *rac*-(2′*R*,3a*S*,4′*S*,5′*R*,6*R*,9a*R*)-1,1′,3-trimethyl-2,2″,7-trioxo-3a,9a-diphenyl-5′-propyl-1,2,3,3a,9,9a-hexahydro-7*H*-dispiro[imidazo[4,5-*e*]thiazolo[3,2-*b*][1,2,4]triazine-6,3′-pyrrolidine-2′,3″-indoline]-4′-carboxylate (4k).** Yield 214 mg (31%); white powder; mp: 190–192 °C. IR (KBr): ν 3156 (NH), 3086, 3033 (Ar), 2957, 2932, 2873 (Alk), 1715, 1647, 1620, 1585 (C=O, C=N) cm^−1^. ^1^H NMR (300 MHz, DMSO-*d*_6_): δ 0.94 (t, *J* = 7.2 Hz, 3H, CH_3_), 1.21 (t, *J* = 7.1 Hz, 3H, CH_3_), 1.30–1.47 (m, 2H, CH_2_), 1.48–1.61 (m, 1H, CH_2_), 1.68–1.76 (m, 1H, CH_2_), 1.98 (s, 3H, 1′-NCH_3_), 2.49 (s, 3H, NCH_3_), 2.61 (s, 3H, NCH_3_), 3.60 (d, *J* = 8.4 Hz, 1H, 5′-H), 4.05 (td, *J* = 8.1, 3.2 Hz, 1H, 4′-H), 4.11–4.24 (m, 2H, OCH_2_), 6.22 (d, *J* = 7.4 Hz, 2H, Ph-2,6), 6.61 (d, *J* = 7.4 Hz, 2H, Ph-2,6), 6.90–7.18 (m, 8H, 2Ph-3-5, 5″-H, 7″-H), 7.29 (d, *J* = 7.1 Hz, 1H, 4″-H), 7.48 (t, *J* = 7.6 Hz, 1H, 6″-H), 7.79 (s, 1H, 9-NH), 10.75 (s, 1H, 1″-NH). ^13^C NMR (75 MHz, DMSO-*d*_6_): δ 13.97, 14.42 (2CH_3_), 17.86 (CH_2_), 25.10, 25.86 (2NCH_3_), 33.18, 34.06 (1′-NCH_3_, CH_2_), 55.11, 55.17 (C-4′, C-5′), 60.90 (OCH_2_), 61.98 (C-6), 79.38, 80.15, 82.79 (C-2′, C-3a, C-9a), 109.83 (C-7″), 122.19, 123.10, 126.72, 127.00, 127.16, 127.33, 127.62, 127.69, 128.14, 130.66 (2Ph-2-6, C-3a″, C-4″, C-5″, C-6″), 133.83, 134.56 (2Ph-1), 143.88, 147.28 (C-7a″, 4a-C=N), 159.09 (2-C=O), 166.28 (7-C=O), 168.78 (COOEt), 175.65 (2″-C=O). HRMS (ESI): *m*/*z* [*M* + H]^+^ calcd for C_37_H_39_N_7_O_5_S: 694.2806; found: 694.2801.

**Mixture of ethyl (2′*R*,3a*S*,4′*S*,6*R*,9a*R*)-1,3-dimethyl-2,2″,7-trioxo-3a,9a-diphenyl-1′-((*R*)-1-phenylethyl)-1,2,3,3a,9,9a-hexahydro-7*H*-dispiro[imidazo[4,5-*e*]thiazolo[3,2-*b*][1,2,4]triazine-6,3′-pyrrolidine-2′,3″-indoline]-4′-carboxylate and ethyl (2′*S*,3a*R*,4′*R*,6*S*,9a*S*)-1,3-dimethyl-2,2″,7-trioxo-3a,9a-diphenyl-1′-((*R*)-1-phenylethyl)-1,2,3,3a,9,9a-hexahydro-7*H*-dispiro[imidazo[4,5-*e*]thiazolo[3,2-*b*][1,2,4]triazine-6,3′-pyrrolidine-2′,3″-indoline]-4′-carboxylate (4l and 4m).** Yield 304 mg (41%); white powder; mp: 205–212 °C. IR (KBr): ν 3155 (NH), 3084,3060, 3031 (Ar), 2974, 2936, 2884 (Alk), 1749, 1720, 1697, 1646, 1619, 1584 (C=O, C=N) cm^−1^. ^1^H NMR (300 MHz, DMSO-*d*_6_) δ 0.91 (d, *J* = 6.7 Hz, 3H, CH_3_), 1.17–1.24 (m, 6H, 2CH_3_), 1.29 (d, *J* = 6.7 Hz, 3H, CH_3_), 2.52 (s, 6H, 2NCH_3_), 2.61 (s, 3H, NCH_3_), 2.62 (s, 3H, NCH_3_), 3.24 (t, *J* = 8.7 Hz, 1H, 5′-H), 3.51 (t, *J* = 8.8 Hz, 1H, 5′-H), 3.60–3.67 (m, 2H, 5′-H, 5′-H), 3.64 (q, *J* = 7.0 Hz, 1H, 1′-NCH), 3.96 (q, *J* = 8.3 Hz, 2H, OCH_2_), 4.05–4.20 (m, 5H, 1′-NCH, OCH_2_, 4′-H, 4′-H), 6.20–6.26 (m, 4H, 2Ph-2,6), 6.64 (d, *J* = 7.6 Hz, 4H, 2Ph-2,6), 6.96–7.34 (m, 26H, 4Ph-3-5, 2Ph, 5″-H, 5″-H, 7″-H, 7″-H), 7.41–7.57 (m, 4H, 4″-H, 4″-H, 6″-H, 6″-H), 7.82 (s, 1H, 9-NH), 7.88 (s, 1H, 9-NH), 10.71 (s, 1H, 1″-NH), 10.84 (s, 1H, 1″-NH). HRMS (ESI): *m*/*z* [*M* + H]^+^ calcd for C_41_H_39_N_7_O_5_S: 742.2806; found: 742.2790.

### 3.3. General Procedure for the Synthesis of Compounds ***5a**–**j*** and ***6a**,**b**,**e**–**g**,**i**,**j***

To a stirred suspension of compounds **4a***–***j** (0.5 mmol) in 10 mL of MeOH (for **4a***–***e**) or absolute EtOH (for **4f***–***j**), 0.125 mmol of MeONa (0.024 mL of 30% solution in MeOH for **4a***–***e**) or 0.125 mmol of EtONa (0.047 mL of 21% solution in EtOH for **4f***–***j**) was added. The resulting mixture was refluxed with stirring for 4 h. The precipitates of compounds **5c**,**d**,**h** that formed after cooling the reaction mass were individual diastereomers.

To obtain the target compounds as mixtures of diastereomers **5** and **6**, the solvent was evaporated under reduced pressure, and the dry residue was triturated with a small amount of MeCN. The resulting suspension was filtered, and the filter cake was washed with MeCN and dried at 50 °C.

To obtain the individual diastereomers **5** and **6**, the resulting precipitate was dissolved in boiling MeCN and the resulting solution was left in an open flask to effect slow crystallization of the precipitate. As the volume of the solution decreased, the crystallizing precipitates were filtered, washed with MeCN, and dried. The filtrate was left in an open flask for further crystallization. This procedure was repeated at least 3–4 times. If necessary, the product contaminated with another isomer could be purified via recrystallization from MeCN.

**Methyl *rac*-(2′*R*,3a*S*,4′*R*,6*R*,9a*R*)-1,1′,3-trimethyl-2,2″,7-trioxo-3a,9a-diphenyl-1,2,3,3a,9,9a-hexahydro-7*H*-dispiro[imidazo[4,5-*e*]thiazolo[3,2-*b*][1,2,4]triazine-6,3′-pyrrolidine-2′,3″-indoline]-4′-carboxylate (5a).** Yield 168 mg (53%); white powder; mp: 295–297 °C. IR (KBr): ν 3282, 3180 (NH), 3079, 3058, 3032 (Ar), 2948, 2909, 2863, 2795 (Alk), 1966, 1909 (Ar), 1738, 1711, 1691, 1647, 1584 (C=O, C=N) cm^−1^. ^1^H NMR (300 MHz, DMSO-*d*_6_): δ 2.13 (s, 3H, 1′-NCH_3_), 2.53 (s, 3H, NCH_3_), 2.68 (s, 3H, NCH_3_), 3.47 (dd, *J* = 11.2, 8.7 Hz, 1H, 5′-H), 3.68 (s, 3H, OCH_3_), 3.79 (dd, *J* = 8.6, 6.4 Hz, 1H, 5′-H), 4.44 (dd, *J* = 11.3, 6.3 Hz, 1H, 4′-H), 6.04 (d, *J* = 8.0 Hz, 2H, Ph-2,6), 6.63 (d, *J* = 7.3 Hz, 2H, Ph-2,6), 6.94 (t, *J* = 7.9 Hz, 2H, Ph-3,5), 7.00–7.21 (m, 6H, 2Ph-4, Ph-3,5, 5″-H, 7″-H), 7.25 (d, *J* = 6.8 Hz, 1H, 4″-H), 7.62 (t, *J* = 7.0 Hz, 1H, 6″-H), 7.93 (s, 1H, 9-NH), 10.91 (s, 1H, 1″-NH). ^13^C NMR (75 MHz, DMSO-*d*_6_): δ 25.05, 25.87 (2NCH_3_), 35.68 (1′-NCH_3_), 45.56, 50.75, 52.30 (C-4′, C-5′, OCH_3_), 62.15 (C-6), 78.84, 79.35, 83.28 (C-2′, C-3a, C-9a), 110.42 (C-7″), 121.85, 122.44, 126.95, 127.03, 127.17, 127.63, 127.95, 128.55, 130.67 (2Ph-2-6, C-3a″, C-4″, C-5″, C-6″), 133.72, 134.56 (2Ph-1), 143.93, 146.00 (C-7a″, 4a-C=N), 159.27 (2-C=O), 163.96 (7-C=O), 170.15 (COOMe), 176.28 (2″-C=O). HRMS (ESI): *m*/*z* [*M* + H]^+^ calcd for C_33_H_31_N_7_O_5_S: 638.2180; found: 638.2168.

**Methyl *rac*-(2′*R*,3a*S*,4′*R*,6*R*,9a*R*)-1′-ethyl-1,3-dimethyl-2,2″,7-trioxo-3a,9a-diphenyl-1,2,3,3a,9,9a-hexahydro-7*H*-dispiro[imidazo[4,5-*e*]thiazolo[3,2-*b*][1,2,4]triazine-6,3′-pyrrolidine-2′,3″-indoline]-4′-carboxylate (5b).** Yield 136 mg (50%); white powder; mp: 315–317 °C. IR (KBr): ν 3259, 3200 (NH), 3078, 3058, 3030 (Ar), 2954, 2916, 2871, 2823 (Alk), 1965, 1913 (Ar), 1742, 1718, 1694, 1644, 1583 (C=O, C=N) cm^−1^. ^1^H NMR (300 MHz, DMSO-*d*_6_): δ 0.94 (t, *J* = 7.2 Hz, 3H, CH_3_), 2.20–2.32 (m, 1H, 1′-NCH_2_), 2.33–2.43 (m, 1H, 1′-NCH_2_), 2.49 (s, 3H, NCH_3_), 2.67 (s, 3H, NCH_3_), 3.42 (dd, *J* = 11.0, 8.6 Hz, 1H, 5′-H), 3.66 (s, 3H, OCH_3_), 3.81 (dd, *J* = 8.7, 6.7 Hz, 1H, 5′-H), 4.44 (dd, *J* = 11.1, 6.8 Hz, 1H, 4′-H), 6.03 (d, *J* = 7.6 Hz, 2H, Ph-2,6), 6.62 (d, *J* = 7.1 Hz, 2H, Ph-2,6), 6.92 (t, *J* = 7.8 Hz, 2H, Ph-3,5), 7.00 (d, *J* = 7.7 Hz, 1H, 7″-H), 7.03–7.19 (m, 5H, Ph-3,5, 2Ph-4, 5″-H), 7.25 (d, *J* = 7.5 Hz, 1H, 4″-H), 7.60 (td, *J* = 7.7, 1.4 Hz, 1H, 6″-H), 7.84 (s, 1H, 9-NH), 10.87 (s, 1H, 1″-NH). ^13^C NMR (75 MHz, DMSO-*d*_6_): δ 13.77 (CH_3_), 24.97, 25.86 (2NCH_3_), 43.81 (1′-NCH_2_), 45.07, 50.36, 52.27 (C-4′, C-5′, OCH_3_), 62.02 (C-6), 78.84, 83.30 (C-2′, C-3a, C-9a), 110.30 (C-7″), 122.47, 125.51, 126.92, 127.03, 127.16, 127.55, 127.62, 128.13, 130.56 (2Ph-2-6, C-3a″, C-4″, C-5″, C-6″), 133.72, 134.56 (2Ph-1), 143.75, 146.03 (C-7a″, 4a-C=N), 159.24 (2-C=O), 163.82 (7-C=O), 170.17 (COOMe), 176.66 (2″-C=O). HRMS (ESI): *m*/*z* [*M* + H]^+^ calcd for C_34_H_33_N_7_O_5_S: 652.2337; found: 652.2338.

**Methyl *rac*-(2′*R*,3a*S*,4′*R*,6*R*,9a*R*)-1′-isopropyl-1,3-dimethyl-2,2″,7-trioxo-3a,9a-diphenyl-1,2,3,3a,9,9a-hexahydro-7*H*-dispiro[imidazo[4,5*-e*]thiazolo[3,2-*b*][1,2,4]triazine-6,3′-pyrrolidine-2′,3″-indoline]-4′-carboxylate (5c).** Yield 299 mg (90%); white powder; mp: >300 °C. IR (KBr): ν 3284, 3250 (NH), 3074, 3058, 3034 (Ar), 2981, 2966, 2934, 2871, 2816 (Alk), 1751, 1718, 1692, 1645, 1620, 1601 (C=O, C=N) cm^−1^. ^1^H NMR (300 MHz, DMSO-*d*_6_): δ 0.92 (d, *J* = 6.7 Hz, 3H, CH_3_), 0.96 (d, *J* = 6.4 Hz, 3H, CH_3_), 2.49 (s, 3H, NCH_3_), 2.66 (s, 3H, NCH_3_), 2.68–2.81 (m, 1H, 1′-NCH), 3.60–3.80 (m, 5H, OCH_3_, 5′-H), 4.45 (dd, *J* = 10.5, 7.5 Hz, 1H, 4′-H), 6.03 (d, *J* = 7.4 Hz, 2H, Ph-2,6), 6.62 (d, *J* = 7.4 Hz, 2H, Ph-2,6), 6.93 (t, *J* = 7.7 Hz, 2H, Ph-3,5), 6.98–7.19 (m, 6H, 2Ph-4, Ph-3,5, 5″-H, 7″-H), 7.23 (d, *J* = 6.6 Hz, 1H, 4″-H), 7.60 (t, *J* = 8.3 Hz, 1H, 6″-H), 7.77 (s, 1H, 9-NH), 10.85 (s, 1H, 1″-NH). ^13^C NMR (75 MHz, DMSO-*d*_6_): δ 17.87, 22.14 (2CH_3_), 24.85, 25.82 (2NCH_3_), 44.30, 44.64, 47.16, 52.21 (1′-NCH, C-4′, C-5′, OCH_3_,), 62.23 (C-6), 77.43, 78.80, 83.28 (C-2′, C-3a, C-9a), 110.43 (C-7″), 122.32, 125.50, 126.88, 127.02, 127.10, 127.49, 128.08, 130.41 (2Ph-2-6, C-3a″, C-4″, C-5″, C-6″), 133.71, 134.57 (2Ph-1), 143.44, 146.09 (C-7a″, 4a-C=N), 159.20 (2-C=O), 163.65 (7-C=O), 170.15 (COOMe), 179.07 (2″-C=O). HRMS (ESI): *m*/*z* [*M* + H]^+^ calcd for C_35_H_35_N_7_O_5_S: 666.2493; found: 666.2482.

**Methyl *rac*-(2′*R*,3a*S*,4′*R*,6*R*,9a*R*)-5″-bromo-1′-ethyl-1,3-dimethyl-2,2″,7-trioxo-3a,9a-diphenyl-1,2,3,3a,9,9a-hexahydro-7*H*-dispiro[imidazo[4,5-*e*]thiazolo[3,2-*b*][1,2,4]triazine-6,3′-pyrrolidine-2′,3″-indoline]-4′-carboxylate (5d).** Yield 295 mg (81%); white powder; mp: >300 °C. IR (KBr): ν 3252 (NH), 3059, 3034 (Ar), 2981, 2943, 2880, 2840 (Alk), 1733, 1690, 1640, 1585 (C=O, C=N) cm^−1^. ^1^H NMR (300 MHz, DMSO-*d*_6_): δ 0.95 (t, *J* = 7.1 Hz, 3H, CH_3_), 2.19–2.31 (m, 1H, 1′-NCH_2_), 2.35–2.44 (m, 1H, 1′-NCH_2_), 2.50 (s, 3H, NCH_3_), 2.67 (s, 3H, NCH_3_), 3.41 (dd, *J* = 11.2, 8.9 Hz, 1H, 5′-H), 3.66 (s, 3H, OCH_3_), 3.82 (dd, *J* = 8.6, 7.2 Hz, 1H, 5′-H), 4.41 (dd, *J* = 11.2, 6.6 Hz, 1H, 4′-H), 6.04 (d, *J* = 7.4 Hz, 2H, Ph-2,6), 6.76 (d, *J* = 7.5 Hz, 2H, Ph-2,6), 6.96–7.14 (m, 7H, 2Ph-3-5, 7″-H), 7.30 (d, *J* = 1.9 Hz, 1H, 4″-H), 7.83 (dd, *J* = 8.3, 2.0 Hz, 1H, 6″-H), 8.30 (s, 1H, 9-NH), 11.05 (s, 1H, 1″-NH). ^13^C NMR (75 MHz, DMSO-*d*_6_): δ 13.81 (CH_3_), 25.02, 25.98 (2NCH_3_), 43.96, 44.73, 50.45, 52.38 (1′-NCH_2,_ C-4′, C-5′, OCH_3_), 63.23 (C-6), 78.94, 79.07, 83.82 (C-2′, C-3a, C-9a), 112.40, 114.64 (C-7″, 5″-CBr), 124.89, 126.88, 127.21, 127.33, 127.52, 127.67, 128.10 (2-Ph-2-6, C-3a″, C-4″), 133.59, 133.98, 134.72 (2Ph-1, 6″-C), 143.18, 145.55 (C-7a″, 4a-C=N), 159.26 (2-C=O), 163.37 (7-C=O), 170.06 (COOMe), 176.23 (2″-C=O). HRMS (ESI): *m*/*z* [*M* + H]^+^ calcd for C_34_H_32_BrN_7_O_5_S: 730.1442; found 730.1419.

**Methyl *rac*-(2′*R*,3a*S*,4′*R*,6*R*,9a*R*)-6″-chloro-1′-ethyl-1,3-dimethyl-2,2″,7-trioxo-3a,9a-diphenyl-1,2,3,3a,9,9a-hexahydro-7*H*-dispiro[imidazo[4,5-*e*]thiazolo[3,2-*b*][1,2,4]triazine-6,3′-pyrrolidine-2′,3″-indoline]-4′-carboxylate (5e).** Yield 65 mg (19%); white powder; mp: 305–307 °C. IR (KBr): ν 3272, 3197 (NH), 3064, 3031 (Ar), 2971, 2950, 2873 (Alk), 1725, 1643, 1614 (C=O, C=N) cm^−1^. ^1^H NMR (300 MHz, DMSO-*d*_6_): δ 0.94 (t, *J* = 7.2 Hz, 3H, CH_3_), 2.21–2.29 (m, 1H, 1′-NCH_2_), 2.34–45 (m, 1H, 1′-NCH_2_), 2.50 (s, 3H, NCH_3_), 2.66 (s, 3H, NCH_3_), 3.41 (dd, *J* = 11.1, 8.8 Hz, 1H, 5′-H), 3.67 (s, 3H, OCH_3_), 3.80 (dd, *J* = 8.8, 7.0 Hz, 1H, 5′-H), 4.45 (dd, *J* = 11.3, 6.6 Hz, 1H, 4′-H), 6.12 (d, *J* = 7.6 Hz, 2H, Ph-2,6), 6.65 (d, *J* = 7.2 Hz, 2H, Ph-2,6), 6.98–7.19 (m, 7H, 2Ph-3-5, 5″-H), 7.24 (s, 2H, 4″-H, 7″-H), 7.81 (s, 1H, 9-NH), 11.07 (s, 1H, 1″-NH). ^13^C NMR (150 MHz, DMSO-*d*_6_): δ 13.87 (CH_3_), 25.10, 25.94 (2NCH_3_), 43.93, 45.14, 50.42, 52.45 (1′-NCH_2_, C-4′, C-5′, OCH_3_), 61.87 (C-6), 78.58, 79.00, 83.31 (C-2′, C-3a, C-9a), 110.54 (C-7″), 121.62, 122.38, 127.01, 127.10, 127.20, 127.35, 127.71, 127.88, 128.30 (2Ph-2-6, C-3a″, C-4″, C-5″), 133.62, 134.56, 135.20 (2Ph-1, 6″-CCl), 145.42, 146.13 (C-7a″, 4a-C=N), 159.30 (2-C=O), 163.84 (7-C=O), 170.15 (COOMe), 176.67 (2″-C=O). HRMS (ESI): *m*/*z* [*M* + H]^+^ calcd for C_34_H_32_ClN_7_O_5_S: 686.1947; found: 686.1935.

**Ethyl *rac*-(2′*R*,3a*S*,4′*R*,6*R*,9a*R*)-1,1′,3-trimethyl-2,2″,7-trioxo-3a,9a-diphenyl-1,2,3,3a,9,9a-hexahydro-7*H*-dispiro[imidazo[4,5-*e*]thiazolo[3,2-*b*][1,2,4]triazine-6,3′-pyrrolidine-2′,3″-indoline]-4′-carboxylate (5f).** Yield 88 mg (27%); white powder; mp: 260–262 °C. IR (KBr): ν 3161, 3088 (NH), 3030 (Ar), 2987, 2941, 2910, 2868, 2790 (Alk), 1729, 1707, 1648, 1584 (C=O, C=N) cm^−1^. ^1^H NMR (300 MHz, DMSO-*d*_6_): δ 1.23 (t, *J* = 7.1 Hz, 3H, CH_3_), 2.13 (s, 3H, 1′-NCH_3_), 2.50 (s, 3H, NCH_3_), 2.67 (s, 3H, NCH_3_), 3.44 (t, *J* = 10.0 Hz, 1H, 5′-H), 3.79 (t, *J* = 7.6 Hz, 1H, 5′-H), 4.02–4.10 (m, 1H, OCH_2_), 4.17–4.26 (m, 1H, OCH_2_), 4.41 (dd, *J* = 11.3, 6.5 Hz, 1H, 4′-H), 6.03 (d, *J* = 8.0 Hz, 2H, Ph-2,6), 6.63 (d, *J* = 7.5 Hz, 2H, Ph-2,6), 6.93 (t, *J* = 7.7 Hz, 2H, Ph-3,5), 6.99–7.20 (m, 6H, 2Ph-4, Ph-3,5, 5″-H, 7″-H), 7.25 (d, *J* = 7.4 Hz, 1H, 4″-H), 7.62 (t, *J* = 7.7 Hz, 1H, 6″-H), 7.90 (s, 1H, 9-NH), 10.90 (s, 1H, 1″-NH). ^13^C NMR (75 MHz, DMSO-*d*_6_): δ 13.87 (CH_3_), 25.01, 25.88 (2NCH_3_), 35.71 (1′-NCH_3_), 45.48, 52.40 (C-4′, C-5′), 60.99 (OCH_2_), 62.19 (C-6), 78.80, 79.27, 83.22 (C-2′, C-3a, C-9a), 110.36 (C-7″), 121.88, 122.44, 125.52, 126.95, 127.02, 127.15, 127.51, 127.60, 128.09, 130.67 (2Ph-2-6, C-3a″, C-4″, C-5″, C-6″), 133.76, 134.62 (2Ph-1), 143.84, 145.98 (C-7a″, 4a-C=N), 159.22 (2-C=O), 163.89 (7-C=O), 169.58 (COOEt), 176.29 (2″-C=O). HRMS (ESI): *m*/*z* [*M* + H]^+^ calcd for C_34_H_33_N_7_O_5_S: 652.2337; found: 652.2327.

**Ethyl *rac*-(2′*R*,3a*S*,4′*R*,6*R*,9a*R*)-1′-ethyl-1,3-dimethyl-2,2″,7-trioxo-3a,9a-diphenyl-1,2,3,3a,9,9a-hexahydro-7*H*-dispiro[imidazo[4,5-*e*]thiazolo[3,2-*b*][1,2,4]triazine-6,3′-pyrrolidine-2′,3″-indoline]-4′-carboxylate (5g).** Yield 173 mg (52%); white powder; mp: >300 °C. IR (KBr): ν 3178, 3092 (NH), 3034 (Ar), 2973, 2936, 2913, 2866, 2836 (Alk), 1746, 1722, 1632, 1585 (C=O, C=N) cm^−1^. ^1^H NMR (300 MHz, DMSO-*d*_6_): δ 0.94 (t, *J* = 7.2 Hz, 3H, CH_3_), 1.22 (t, *J* = 7.1 Hz, 3H, CH_3_), 2.21–2.28 (m, 1H, 1′ -NCH_2_), 2.36–2.44 (m, 1H, 1′ -NCH_2_), 2.49 (s, 3H, NCH_3_), 2.66 (s, 3H, NCH_3_), 3.41 (dd, *J* = 11.4, 8.6 Hz, 1H, 5′-H), 3.82 (t, *J* = 7.7 Hz, 1H, 5′-H), 4.03–4.11 (m, 1H, OCH_2_), 4.15–4.23 (m, 1H, OCH_2_), 4.42 (dd, *J* = 11.2, 6.8 Hz, 1H, 4′-H), 6.02 (d, *J* = 7.4 Hz, 2H, Ph-2,6), 6.62 (d, *J* = 7.1 Hz, 2H, Ph-2,6), 6.90–7.17 (m, 8H, 2Ph-3-5, 5″-H, 7″-H), 7.26 (d, *J* = 7.7 Hz, 1H, 4″-H), 7.60 (t, *J* = 7.7 Hz, 1H, 6″-H), 7.84 (s, 1H, 9-NH), 10.87 (s, 1H, 1″-NH). ^13^C NMR (150 MHz, DMSO-*d*_6_): δ 13.83, 13.93 (2CH_3_), 24.99, 25.95 (2NCH_3_), 43.93 (1′-NCH_2_), 45.04, 50.37 (C-4′, C-5′), 61.14 (OCH_2_), 62.16 (C-6), 78.90, 83.34 (C-2′, C-3a, C-9a), 110.42 (C-7″), 122.55, 122.58, 125.57, 126.95, 127.07, 127.24, 127.65, 127.74, 128.26, 130.67 (2Ph-2-6, C-3a″, C-4″, C-5″, C-6″), 133.70, 134.58 (2Ph-1), 143.77, 146.16 (C-7a″, 4a-C=N), 159.38 (2-C=O), 163.89 (7-C=O), 169.74 (COOEt), 176.87 (2″-C=O). HRMS (ESI): *m*/*z* [*M* + H]^+^ calcd for C_35_H_35_N_7_O_5_S: 666.2493; found: 666.2491.

**Ethyl *rac*-(2′*R*,3a*S*,4′*R*,6*R*,9a*R*)-1′-isopropyl-1,3-dimethyl-2,2″,7-trioxo-3a,9a-diphenyl-1,2,3,3a,9,9a-hexahydro-7*H*-dispiro[imidazo[4,5-*e*]thiazolo[3,2-*b*][1,2,4]triazine-6,3′-pyrrolidine-2′,3″-indoline]-4′-carboxylate (5h).** Yield 247 mg (73%); white powder; mp: >300 °C. IR (KBr): ν 3292, 3259 (NH), 3085, 3059, 3036 (Ar), 2964, 2933, 2872, 2816 (Alk), 1747, 1719, 1693, 1642, 1622, 1602 (C=O, C=N) cm^−1^. ^1^H NMR (300 MHz, DMSO-*d*_6_): δ 0.92–0.98 (m, 6H, 2CH_3_), 1.23 (t, *J* = 7.1 Hz, 3H, CH_3_), 2.49 (s, 3H, NCH_3_), 2.65 (s, 3H, NCH_3_), 2.74–2.78 (m, 1H, 1′-NCH), 3.66–3.74 (m, 2H, 5′-H, 5′-H), 4.06–4.23 (m, 2H, OCH_2_), 4.44 (t, *J* = 9.0 Hz, 1H, 4′-H), 6.03 (d, *J* = 8.2 Hz, 2H, Ph-2,6), 6.62 (d, *J* = 7.5 Hz, 2H, Ph-2,6), 6.94 (t, *J* = 7.8 Hz, 2H, Ph-3,5), 6.99–7.17 (m, 6H, Ph-3,5, 2Ph-4, 5″-H, 7″-H), 7.24 (d, *J* = 7.1 Hz, 1H, 4″-H), 7.61 (t, *J* = 7.4 Hz, 1H, 6″-H), 7.78 (s, 1H, 9-NH), 10.88 (s, 1H, 1″-NH). ^13^C NMR (75 MHz, DMSO-*d*_6_): δ 13.86, 17.89, 22.14 (3CH_3_), 24.82, 25.82 (2NCH_3_), 44.25, 44.60, 47.15 (1′-NCH, C-4′, C-5′), 60.90 (OCH_2_), 62.28 (C-6), 77.39, 78.76, 83.23 (C-2′, C-3a, C-9a), 110.45 (C-7″), 121.84, 122.27, 122.92, 125.46, 126.88, 127.01, 127.09, 127.47, 128.06, 130.38 (2Ph-2-6, C-3a″, C-4″, C-5″, C-6″), 133.75, 134.61 (2Ph-1), 143.47, 146.10 (C-7a″, 4a-C=N), 159.17 (2-C=O), 163.60 (7-C=O), 169.60 (COOEt), 179.08 (2″-C=O). HRMS (ESI): *m*/*z* [*M* + H]^+^ calcd for C_36_H_37_N_7_O_5_S: 680.2650; found: 680.2636.

**Ethyl *rac*-(2′*R*,3a*S*,4′*R*,6*R*,9a*R*)-5″-bromo-1′-ethyl-1,3-dimethyl-2,2″,7-trioxo-3a,9a-diphenyl-1,2,3,3a,9,9a-hexahydro-7*H*-dispiro[imidazo[4,5-*e*]thiazolo[3,2-*b*][1,2,4]triazine-6,3′-pyrrolidine-2′,3″-indoline]-4′-carboxylate (5i).** Yield 52 mg (14%); white powder; mp: 194–197 °C. IR (KBr): ν 3248, 3187 (NH), 3091, 3064, 3034 (Ar), 2975, 2937, 2874, 2828 (Alk), 1957, 1890 (Ar), 1723, 1645, 1585 (C=O, C=N) cm^−1^. ^1^H NMR (300 MHz, DMSO-*d*_6_): δ 0.97 (t, *J* = 7.2 Hz, 3H, CH_3_), 1.23 (t, *J* = 7.1 Hz, 3H, CH_3_), 2.24–2.31 (m, 1H, 1′-NCH_2_), 2.41–2.47 (m, 1H, 1′-NCH_2_), 2.51 (s, 3H, NCH_3_), 2.68 (s, 3H, NCH_3_), 3.42 (dd, *J* = 11.3, 8.7 Hz, 1H, 5′-H), 3.83 (t, *J* = 7.7 Hz, 1H, 5′-H), 4.04–4.10 (m, 1H, OCH_2_), 4.18–4.24 (m, 1H, OCH_2_), 4.40 (dd, *J* = 11.3, 6.8 Hz, 1H, 4′-H), 6.06 (d, *J* = 7.7 Hz, 2H, Ph-2,6), 6.78 (d, *J* = 7.6 Hz, 2H, Ph-2,6), 6.90–7.20 (m, 7H, 2Ph-3-5, 7″-H), 7.33 (d, *J* = 2.2 Hz, 1H, 4″-H), 7.85 (dd, *J* = 8.3, 2.2 Hz, 1H, 6″-H), 8.31 (s, 1H, 9-NH), 11.06 (s, 1H, 1″-NH). ^13^C NMR (75 MHz, DMSO-*d*_6_): δ 13.80, 13.84 (2CH_3_), 24.98, 25.94 (2NCH_3_), 43.93, 44.62, 50.36 (1′-NCH_2_, C-4′, C-5′), 61.07 (OCH_2_), 62.25 (C-6), 78.85, 78.97, 83.67 (C-2′, C-3a, C-9a), 112.32, 114.56 (5″-CBr, C-7″), 124.91, 126.81, 127.12, 127.43, 127.56, 128.04 (2Ph-2-6, C-3a″, C-4″), 133.51, 133.95, 134.74 (2Ph-1, C-6″), 143.10, 145.50 (C-7a″, 4a-C=N), 159.15 (2-C=O), 163.26 (7-C=O), 169.42 (COOEt), 176.21 (2″-C=O). HRMS (ESI): *m*/*z* [*M* + H]^+^ calcd for C_35_H_34_BrN_7_O_5_S: 744.1598; found: 744.1595.

**Ethyl *rac*-(2′*R*,3a*S*,4′*R*,6*R*,9a*R*)-6″-chloro-1′-ethyl-1,3-dimethyl-2,2″,7-trioxo-3a,9a-diphenyl-1,2,3,3a,9,9a-hexahydro-7*H*-dispiro[imidazo[4,5-*e*]thiazolo[3,2-*b*][1,2,4]triazine-6,3′-pyrrolidine-2′,3″-indoline]-4′-carboxylate (5j).** Yield 168 mg (48%); white powder; mp: 192–195 °C. IR (KBr): ν 3162, 3064 (NH), 3034 (Ar), 2974, 2937, 2874, 2824 (Alk), 1727, 1646, 1616 (C=O, C=N) cm^−1^. ^1^H NMR (300 MHz, DMSO-*d*_6_): δ 0.95 (t, *J* = 7.1 Hz, 3H, CH_3_), 1.23 (t, *J* = 7.1 Hz, 3H, CH_3_), 2.22–2.28 (m, 1H, 1′-NCH_2_), 2.38–2.45 (m, 1H, 1′-NCH_2_), 2.51 (s, 3H, NCH_3_), 2.66 (s, 3H, NCH_3_), 3.41 (t, *J* = 9.9 Hz, 1H, 5′-H), 3.81 (t, *J* = 7.8 Hz, 1H, 5′-H), 4.04–4.11 (m, 1H, OCH_2_), 4.18–4.24 (m, 1H, OCH_2_), 4.43 (dd, *J* = 11.1, 6.8 Hz, 1H, 4′-H), 6.13 (d, *J* = 7.8 Hz, 2H, Ph-2,6), 6.66 (d, *J* = 7.1 Hz, 2H, Ph-2,6), 6.90–7.19 (m, 7H, 2Ph-3-5, 5″-H), 7.25 (s, 2H, 4″-H, 7″-H), 7.82 (s, 1H, 9-NH), 11.07 (s, 1H, 1″-NH). ^13^C NMR (150 MHz, DMSO-*d*_6_): δ 13.79, 13.87 (2CH_3_), 24.94, 25.85 (2NCH_3_), 43.86, 45.04, 50.32 (1′-NCH_2_, C-4′, C-5′), 61.05 (OCH_2_), 61.88 (C-6), 78.42, 78.90, 83.20 (C-2′, C-3a, C-9a), 110.44 (C-7″), 121.61, 122.25, 126.92, 127.02, 127.08, 127.59, 127.75, 128.17 (2Ph-2-6, C-3a″, C-4″, C-5″), 133.60, 134.54, 135.10 (2Ph-1, 6″-CCl), 145.31, 146.03 (C-7a″, 4a-C=N), 159.16 (2-C=O), 163.69 (7-C=O), 169.48 (COOEt), 176.64 (2″-C=O). HRMS (ESI): *m*/*z* [*M* + H]^+^ calcd for C_35_H_34_ClN_7_O_5_S: 700.2103; found: 700.2118.

**Methyl *rac*-(2′*R*,3a*S*,4′*R*,6*S*,9a*R*)-1,1′,3-trimethyl-2,2″,7-trioxo-3a,9a-diphenyl-1,2,3,3a,9,9a-hexahydro-7*H*-dispiro[imidazo[4,5-*e*]thiazolo[3,2-*b*][1,2,4]triazine-6,3′-pyrrolidine-2′,3″-indoline]-4′-carboxylate (6a).** Yield 41 mg (13%); white powder; mp: 255–257 °C. IR (KBr): ν 3367, 3159 (NH), 3063 (Ar), 2979, 2949, 2887, 2800 (Alk), 1969, 1911 (Ar), 1751, 1731, 1701, 1642, 1584 (C=O, C=N) cm^−1^. ^1^H NMR (300 MHz, DMSO-*d*_6_): δ 1.90 (s, 3H, NCH_3_), 2.11 (s, 3H, 1′-NCH_3_), 2.52 (s, 3H, NCH_3_), 3.46 (t, *J* = 10.0 Hz, 1H, 5′-H), 3.64 (t, *J* = 8.2 Hz, 1H, 5′-H), 3.86 (s, 3H, OCH_3_), 5.05 (dd, *J* = 10.7, 7.1 Hz, 1H, 4′-H), 6.53–6.69 (m, 4H, 2Ph-2,6), 6.80 (d, *J* = 7.7 Hz, 1H, 7″-H), 6.90–7.20 (m, 7H, 2Ph-3-5, 5″-H), 7.37 (t, *J* = 8.2 Hz, 1H, 6″-H), 7.63 (d, *J* = 7.5 Hz, 1H, 4″-H), 7.72 (s, 1H, 9-NH), 10.56 (s, 1H, 1″-NH). ^13^C NMR (75 MHz, DMSO-*d*_6_): δ 23.82, 25.83 (2NCH_3_), 34.69 (1′-NCH_3_), 48.59 (C-4′), 50.75 (C-5′), 52.36 (OCH_3_), 67.65 (C-6), 77.77 (C-2′), 79.01 (C-9a), 82.31 (C-3a), 109.52 (C-7″), 121.45 (C-4″), 121.94 (C-3a″), 126.52, 127.21, 127.38, 127.58, 127.78, 128.03 (2Ph-2-6, C-5″), 130.64 (C-6″), 133.85, 134.63 (2Ph-1), 143.21 (C-7a″), 147.21 (4a-C=N), 158.54 (2-C=O), 163.43 (7-C=O), 169.63 (COOMe), 174.58 (2″-C=O). HRMS (ESI): *m*/*z* [*M* + H]^+^ calcd for C_33_H_31_N_7_O_5_S: 638.2180; found: 638.2179.

**Methyl *rac*-(2′*R*,3a*S*,4′*R*,6*S*,9a*R*)-1′-ethyl-1,3-dimethyl-2,2″,7-trioxo-3a,9a-diphenyl-1,2,3,3a,9,9a-hexahydro-7*H*-dispiro[imidazo[4,5-*e*]thiazolo[3,2-*b*][1,2,4]triazine-6,3′-pyrrolidine-2′,3″-indoline]-4′-carboxylate (6b).** Yield 16 mg (5%); white powder; mp: 245–247 °C. IR (KBr): ν 3320, 3212 (NH), 3054, 3031 (Ar), 2973, 2934, 2918, 2883, 2843 (Alk), 1750, 1735, 1717, 1645 (C=O, C=N) cm^−1^. ^1^H NMR (300 MHz, DMSO-*d*_6_): δ 0.98 (t, *J* = 7.2 Hz, 3H, CH_3_), 1.88 (s, 3H, NCH_3_), 2.20–2.26 (m, 1H, 1′-NCH_2_), 2.34–2.41 (m, 1H, 1′-NCH_2_), 2.52 (s, 3H, NCH_3_), 3.41 (t, *J* = 10.0 Hz, 1H, 5′-H), 3.63–3.68 (m, 1H, 5′-H), 3.87 (s, 3H, OCH_3_), 5.07 (dd, *J* = 10.6, 7.2 Hz, 1H, 4′-H), 6.62–6.65 (m, 4H, 2Ph-2,6), 6.78 (d, *J* = 7.7 Hz, 1H, 7″-H), 6.90–7.20 (m, 7H, 2Ph-3-5, 5″-H), 7.36 (t, *J* = 7.6 Hz, 1H, 6″-H), 7.64 (d, *J* = 7.2 Hz, 1H, 4″-H), 7.73 (s, 1H, 9-NH), 10.56 (s, 1H, 1″-NH). ^13^C NMR (75 MHz, DMSO-*d*_6_): δ 13.58 (CH_3_), 23.83, 25.84 (2NCH_3_), 42.93 (1′-NCH_2_), 48.27, 48.54, 52.42 (C-4′, C-5′, OCH_3_), 67.52 (C-6), 77.55, 78.99, 82.26 (C-2′, C-3a, C-9a), 109.49 (C-7″), 121.45, 122.43, 126.52, 127.25, 127.38, 127.62, 127.81, 128.07, 130.60 (2Ph-2-6, C-3a″, C-4″, C-5″, C-6″), 133.84, 134.63 (2Ph-1), 143.18 (C-7a″), 147.24 (4a-C=N), 158.55 (2-C=O), 163.40 (7-C=O), 169.74 (COOMe), 174.95 (2″-C=O). HRMS (ESI): *m*/*z* [*M* + H]^+^ calcd for C_34_H_33_N_7_O_5_S: 652.2337; found: 652.2323.

**Methyl *rac*-(2′*R*,3a*S*,4′*R*,6*S*,9a*R*)-6″-chloro-1′-ethyl-1,3-dimethyl-2,2″,7-trioxo-3a,9a-diphenyl-1,2,3,3a,9,9a-hexahydro-7*H*-dispiro[imidazo[4,5-*e*]thiazolo[3,2-*b*][1,2,4]triazine-6,3′-pyrrolidine-2′,3″-indoline]-4′-carboxylate (6e).** Yield 65 mg (19%); white powder; mp: 269–271 °C. IR (KBr): ν 3305, 3264 (NH), 3061 (Ar), 2972, 2948, 2881, 2835 (Alk), 1734, 1704, 1643, 1618 (C=O, C=N) cm^−1^. ^1^H NMR (300 MHz, DMSO-*d*_6_): δ 0.97 (t, *J* = 7.2 Hz, 3H, CH_3_), 1.95 (s, 3H, NCH_3_), 2.20–2.28 (m, 1H, 1′-NCH_2_), 2.32–2.42 (m, 1H, 1′-NCH_2_), 2.52 (s, 3H, NCH_3_), 3.39 (t, *J* = 10.1 Hz, 1H, 5′-H), 3.64 (t, *J* = 8.2 Hz, 1H, 5′-H), 3.85 (s, 3H, OCH_3_), 5.02 (dd, *J* = 10.3, 7.0 Hz, 1H, 4′-H), 6.61–6.67 (m, 4H, 2Ph-2,6), 6.80 (s, 1H, 7″-H), 7.00–7.19 (m, 7H, 2Ph-3-5, 5″-H), 7.63 (d, *J* = 8.2 Hz, 1H, 4″-H), 7.75 (s, 1H, 9-NH), 10.74 (s, 1H, 1″-NH). ^13^C NMR (150 MHz, DMSO-*d*_6_): δ 13.66 (CH_3_), 23.71, 25.95 (2NCH_3_), 43.06 (1′-NCH_2_), 48.14, 48.59, 52.53 (C-4′, C-5′, OCH_3_), 67.53 (C-6), 77.40, 79.07, 82.19 (C-2′, C-3a, C-9a), 109.69 (C-7″), 121.27, 121.45, 126.57, 127.36, 127.43, 127.74, 127.92, 128.17, 128.68 (2Ph-2-6, C-3a″, C-4″, C-5″), 133.71, 134.57, 135.48 (2Ph-1, 6″-CCl), 144.69 (C-7a″), 147.18 (4a-C=N), 158.59 (2-C=O), 163.41 (7-C=O), 169.68 (COOMe), 174.85 (2″-C=O). HRMS (ESI): *m*/*z* [*M* + H]^+^ calcd for C_34_H_32_ClN_7_O_5_S: 686.1947; found: 686.1944.

**Ethyl *rac*-(2′*R*,3a*S*,4′*R*,6*S*,9a*R*)-1,1′,3-trimethyl-2,2″,7-trioxo-3a,9a-diphenyl-1,2,3,3a,9,9a-hexahydro-7*H*-dispiro[imidazo[4,5-*e*]thiazolo[3,2-*b*][1,2,4]triazine-6,3′-pyrrolidine-2′,3″-indoline]-4′-carboxylate (6f).** Yield 94 mg (29%); white powder; mp: 239–240 °C. IR (KBr): ν 3370, 3165 (NH), 3064 (Ar), 2978, 2943, 2883, 2798 (Alk), 1731, 1699, 1638, 1584 (C=O, C=N) cm^−1^. ^1^H NMR (300 MHz, DMSO-*d*_6_): δ 1.36 (t, *J* = 7.0 Hz, 3H, CH_3_), 1.91 (s, 3H, NCH_3_), 2.11 (s, 3H, 1′-NCH_3_), 2.50 (s, 3H, NCH_3_), 3.43 (t, *J* = 9.7 Hz, 1H, 5′-H), 3.63 (t, *J* = 8.3 Hz, 1H, 5′-H), 4.31 (q, *J* = 7.2 Hz, 2H, OCH_2_), 5.05 (dd, *J* = 10.6, 7.2 Hz, 1H, 4′-H), 6.51–6.68 (m, 4H, 2Ph-2,6), 6.80 (d, *J* = 7.7 Hz, 1H, 7″-H), 6.90–7.20 (m, 7H, 2Ph-3-5, 5″-H), 7.37 (t, *J* = 8.0 Hz, 1H, 6″-H), 7.64 (d, *J* = 7.5 Hz, 1H, 4″-H), 7.88 (s, 1H, 9-NH), 10.59 (s, 1H, 1″-NH). ^13^C NMR (75 MHz, DMSO-*d*_6_): δ 13.97 (CH_3_), 23.84, 25.83 (2NCH_3_), 34.70 (1′-NCH_3_), 48.29, 50.64 (C-4′, C-5′), 61.19 (OCH_2_), 67.72 (C-6), 77.88, 79.20, 82.67 (C-2′, C-3a, C-9a), 109.51 (C-7″), 121.51, 121.88, 126.61, 127.07, 127.28, 127.36, 127.54, 127.76, 128.04, 130.67 (2Ph-2-6, C-3a″, C-4″, C-5″, C-6″), 133.83, 134.81 (2Ph-1), 143.14 (C-7a″), 146.84 (4a-C=N), 158.33 (2-C=O), 162.89 (7-C=O), 169.00 (COOEt), 174.54 (2″-C=O). HRMS (ESI): *m*/*z* [*M* + H]^+^ calcd for C_34_H_33_N_7_O_5_S: 652.2337; found: 652.2336.

**Ethyl *rac*-(2′*R*,3a*S*,4′*R*,6*S*,9a*R*)-1′-ethyl-1,3-dimethyl-2,2″,7-trioxo-3a,9a-diphenyl-1,2,3,3a,9,9a-hexahydro-7*H*-dispiro[imidazo[4,5-*e*]thiazolo[3,2-*b*][1,2,4]triazine-6,3′-pyrrolidine-2′,3″-indoline]-4′-carboxylate (6g).** Yield 10 mg (3%); white powder; mp: 281–283 °C. IR (KBr): ν 3308, 3224 (NH), 3091, 3063, 3035 (Ar), 2981, 2924, 2874, 2826 (Alk), 1728, 1700, 1635, 1583 (C=O, C=N) cm^−1^. ^1^H NMR (300 MHz, DMSO-*d*_6_): δ 0.98 (t, *J* = 7.1 Hz, 3H, CH_3_), 1.36 (t, *J* = 7.1 Hz, 3H, CH_3_), 1.91 (s, 3H, NCH_3_), 2.19–2.28 (m, 1H, 1′ -NCH_2_), 2.34–2.40 (m, 1H, 1′ -NCH_2_), 2.50 (s, 3H, NCH_3_), 3.39 (t, *J* = 10.0 Hz, 1H, 5′-H), 3.65 (dd, *J* = 8.6, 7.8 Hz, 1H, 5′-H), 4.31 (q, *J* = 7.0 Hz, 2H, OCH_2_), 5.07 (dd, *J* = 10.5, 7.3 Hz, 1H, 4′-H), 6.60–6.65 (m, 4H, 2Ph-2,6), 6.78 (d, *J* = 7.7 Hz, 1H, 7″-H), 6.95–7.16 (m, 7H, 2Ph-3-5, 5″-H), 7.36 (d, *J* = 7.6 Hz, 1H, 6″-H), 7.65 (d, *J* = 7.6 Hz, 1H, 4″-H), 7.87 (s, 1H, 9-NH), 10.57 (s, 1H, 1″-NH). ^13^C NMR (75 MHz, DMSO-*d*_6_): δ 13.58, 14.00 (2CH_3_), 23.87, 25.86 (2NCH_3_), 42.92 (1′-NCH_2_), 47.96, 48.45 (C-4′, C-5′), 61.23 (OCH_2_), 67.63 (C-6), 77.68, 79.22, 82.68 (C-2′, C-3a, C-9a), 109.48 (C-7″), 121.52, 122.41, 126.63, 127.11, 127.19, 127.38, 127.57, 127.78, 128.06, 130.63 (2Ph-2-6, C-3a″, C-4″, C-5″, C-6″), 133.86, 134.86 (2Ph-1), 143.14 (C-7a″), 146.87 (4a-C=N), 158.35 (2-C=O), 162.85 (7-C=O), 169.11 (COOEt), 174.93 (2″-C=O). HRMS (ESI): *m*/*z* [*M* + H]^+^ calcd for C_35_H_35_N_7_O_5_S: 666.2493; found: 666.2503.

**Ethyl *rac*-(2′*R*,3a*S*,4′*R*,6*S*,9a*R*)-5″-bromo-1′-ethyl-1,3-dimethyl-2,2″,7-trioxo-3a,9a-diphenyl-1,2,3,3a,9,9a-hexahydro-7*H*-dispiro[imidazo[4,5-*e*]thiazolo[3,2-*b*][1,2,4]triazine-6,3′-pyrrolidine-2′,3″-indoline]-4′-carboxylate (6i).** Yield 119 mg (32%); white powder; mp: 292–293 °C. IR (KBr): ν 3272, 3150 (NH), 3048 (Ar), 2975, 2936, 2913, 2885, 2836 (Alk), 1753, 1733, 1699, 1636, 1584 (C=O, C=N) cm^−1^. ^1^H NMR (300 MHz, DMSO-*d*_6_): δ 0.95 (t, *J* = 7.2 Hz, 3H, CH_3_), 1.32 (t, *J* = 7.1 Hz, 3H, CH_3_), 2.00 (s, 3H, NCH_3_), 2.15–2.22 (m, 1H, 1′-NCH_2_), 2.31–2.37 (m, 1H, 1′-NCH_2_), 2.48 (s, 3H, NCH_3_), 3.34 (t, *J* = 9.9 Hz, 1H, 5′-H), 3.61 (t, *J* = 8.2 Hz, 1H, 5′-H), 4.19–4.35 (m, 2H, OCH_2_), 4.98 (dd, *J* = 10.7, 7.1 Hz, 1H, 4′-H), 6.58–6.62 (m, 4H, 2Ph-2,6), 6.73 (d, *J* = 8.3 Hz, 1H, 7″-H), 6.96–7.12 (m, 6H, 2Ph-3-5), 7.48 (d, *J* = 8.5 Hz, 1H, 6″-H), 7.66 (s, 1H, 4″-H), 7.83 (s, 1H, 9-NH), 10.67 (br.s., 1H, 1″-NH). ^13^C NMR (75 MHz, DMSO-*d*_6_): δ 13.59, 13.98 (2CH_3_), 24.24, 26.03 (2NCH_3_), 42.99 (1′-NCH_2_), 47.67, 48.42 (C-4′, C-5′), 61.17 (OCH_2_), 67.53 (C-6), 77.91, 79.36, 82.34 (C-2′, C-3a, C-9a), 111.45 (C-7″), 113.34, 124.67, 126.56, 127.10, 127.26, 127.57, 127.67, 127.99, 129.87 (2Ph-2-6, C-3a″, C-4″, C-5″), 133.58, 133.90, 135.16 (2Ph-1, C-6″), 142.67 (C-7a″), 146.52 (4a-C=N), 158.11 (2-C=O), 162.64 (7-C=O), 168.92 (COOEt), 174.65 (2″-C=O). HRMS (ESI): *m*/*z* [*M* + H]^+^ calcd for C_35_H_34_BrN_7_O_5_S: 744.1598; found: 744.159.

**Ethyl *rac*-(2′*R*,3a*S*,4′*R*,6*S*,9a*R*)-6″-chloro-1′-ethyl-1,3-dimethyl-2,2″,7-trioxo-3a,9a-diphenyl-1,2,3,3a,9,9a-hexahydro-7*H*-dispiro[imidazo[4,5-*e*]thiazolo[3,2-*b*][1,2,4]triazine-6,3′-pyrrolidine-2′,3″-indoline]-4′-carboxylate (6j).** Yield 39 mg (11%); white powder; mp: 280–282 °C. IR (KBr): ν 3300 (NH), 3065 (Ar), 2974, 2939, 2878 (Alk), 1734, 1701, 1637, 1616 (C=O, C=N) cm^−1^. ^1^H NMR (300 MHz, DMSO-*d*_6_): δ 0.98 (t, *J* = 7.2 Hz, 3H, CH_3_), 1.36 (t, *J* = 7.1 Hz, 3H, CH_3_), 1.98 (s, 3H, NCH_3_), 2.09–2.26 (m, 1H, 1′-NCH_2_), 2.32–2.40 (m, 1H, 1′-NCH_2_), 2.52 (s, 3H, NCH_3_), 3.38 (t, *J* = 9.8 Hz, 1H, 5′-H), 3.65 (t, *J* = 8.2 Hz, 1H, 5′-H), 4.25–4.37 (m, 2H, OCH_2_), 5.05 (dd, *J* = 10.8, 7.2 Hz, 1H, 4′-H), 6.63–6.64 (m, 4H, 2Ph-2,6), 6.81 (s, 1H, 7″-H), 6.92–7.20 (m, 7H, 2Ph-3-5, 5″-H), 7.64 (d, *J* = 8.2 Hz, 1H, 4″-H), 7.89 (s, 1H, 9-NH). ^13^C NMR (75 MHz, DMSO-*d*_6_): δ 12.48, 12.89 (2CH_3_), 22.56, 24.78 (2NCH_3_), 41.88 (1′-NCH_2_), 46.64, 47.32 (C-4′, C-5′), 60.15 (OCH_2_), 66.41 (C-6), 76.38, 78.09, 81.44 (C-2′, C-3a, C-9a), 108.60 (C-7″), 120.14, 120.19, 125.49, 126.03, 126.23, 126.51, 126.70, 126.98, 127.28 (2Ph-2-6, C-3a″, C-4″, C-5″), 132.53, 133.61, 134.29 (2Ph-1, 6″-CCl), 144.10, 145.67 (C-7a″, 4a-C=N), 157.23 (2-C=O), 161.68 (7-C=O), 167.90 (COOEt), 173.91 (2″-C=O). HRMS (ESI): *m*/*z* [*M* + H]^+^ calcd for C_35_H_34_ClN_7_O_5_S: 700.2103; found: 700.2107.

### 3.4. General Procedure for the Synthesis of Compounds ***11a**–**f***

A mixture of corresponding compound **10a**,**b** (0.5 mmol), amino acetic acid **7a**,**b** (0.75 mmol) and isatin **8a***–***c** (0.75 mmol) in MeCN (15 mL) was refluxed with stirring for 12 h. After cooling, the precipitate of compounds **11a***–***f** was filtered off, washed with methanol and dried at 50 °C.

**Methyl *rac*-(2′*R*,3a*S*,4′*R*,7*S*,9a*R*)-1,1′,3-trimethyl-2,2″,8-trioxo-3a,9a-diphenyl-1,2,3,3a,4,9a-hexahydro-8*H*-dispiro[imidazo[4,5-*e*]thiazolo[2,3-*c*][1,2,4]triazine-7,3′-pyrrolidine-2′,3″-indoline]-4′-carboxylate (11a).** Yield 162 mg (51%); white powder; mp: 258–260 °C. IR (KBr): ν 3335, 3228 (NH), 3065, 3033 (Ar), 2955, 2927, 2877, 2796 (Alk), 1750, 1729, 1704, 1650, 1622, 1599 (C=O, C=N) cm^−1^. ^1^H NMR (300 MHz, DMSO-*d*_6_): δ 2.03 (s, 3H, 1′-NCH_3_), 2.33 (s, 3H, NCH_3_), 2.38 (s, 3H, NCH_3_), 3.29–3.35 (m, 1H, 5′-H), 3.54 (t, *J* = 8.1 Hz, 1H, 5′-H), 3.82 (s, 3H, OCH_3_), 4.78 (dd, *J* = 10.7, 6.9 Hz, 1H, 4′-H), 6.47–6.68 (m, 2H, Ph-2,6), 6.75–7.34 (m, 11H, Ph-2,6, 2Ph-3-5, 5″-H, 6″-H, 7″-H), 7.62 (d, *J* = 7.5 Hz, 1H, 4″-H), 7.73 (s, 1H, 4-NH), 10.44 (s, 1H, 1″-NH). ^13^C NMR (75 MHz, DMSO-*d*_6_): δ 25.20, 29.37 (2NCH_3_), 34.52 (1′-NCH_3_), 48.71, 50.67, 52.23 (C-4′, C-5′, OCH_3_), 68.89 (C-7), 77.82, 81.27, 86.46 (C-2′, C-3a, C-9a), 110.12 (C-7″), 120.93, 121.70, 126.81, 127.25, 127.67, 127.75, 128.02, 128.51, 130.81 (2Ph-2-6, C-3a″, C-4″, C-5″, C-6″), 132.14, 133.31 (2Ph-1), 137.06 (5a-C=N), 142.97 (C-7a″), 157.12 (2-C=O), 167.40, 169.55 (8-C=O, COOMe), 174.84 (2″-C=O). HRMS (ESI): *m*/*z* [*M* + H]^+^ calcd for C_33_H_31_N_7_O_5_S: 638.2180; found: 638.2194.

**Methyl *rac*-(2′*R*,3a*S*,4′*R*,7*S*,9a*R*)-1′-ethyl-1,3-dimethyl-2,2″,8-trioxo-3a,9a-diphenyl-1,2,3,3a,4,9a-hexahydro-8*H*-dispiro[imidazo[4,5-*e*]thiazolo[2,3-*c*][1,2,4]triazine-7,3′-pyrrolidine-2′,3″-indoline]-4′-carboxylate (11b).** Yield 176 mg (54%); white powder; mp: 234–235 °C. IR (KBr): ν 3312, 3268 (NH), 3094, 3065 (Ar), 2974, 2951, 2909, 2895, 2872 (Alk), 1962 (Ar), 1724, 1659, 1616 (C=O, C=N) cm^−1^. ^1^H NMR (300 MHz, DMSO-*d*_6_): δ 0.95 (t, *J* = 7.0 Hz, 3H, CH_3_), 2.12–2.21 (m, 1H, 1′-NCH_2_), 2.24–2.30 (m, 1H, 1′-NCH_2_), 2.35 (s, 3H, NCH_3_), 2.40 (s, 3H, NCH_3_), 3.26–3.35 (m, 1H, 5′-H), 3.59 (t, *J* = 7.4 Hz, 1H, 5′-H), 3.84 (s, 3H, OCH_3_), 4.82 (t, *J* = 9.2 Hz, 1H, 4′-H), 6.55–6.57 (m, 2H, Ph-2,6), 6.76–7.35 (m, 11H, Ph-2,6, 2Ph-3-5, 5″-H, 6″-H, 7″-H), 7.64 (d, *J* = 7.9 Hz, 1H, 4″-H), 7.74 (s, 1H, 4-NH), 10.43 (s, 1H, 1″-NH). ^13^C NMR (75 MHz, DMSO-*d*_6_): δ 13.49 (CH_3_), 25.22, 29.39 (2NCH_3_), 42.69 (1′-NCH_2_), 48.37, 52.26 (C-4′, C-5′, OCH_3_), 68.72 (C-7), 77.66, 81.26, 86.41 (C-2′, C-3a, C-9a), 110.09 (C-7″), 120.93, 122.20, 126.83, 127.25, 127.68, 128.03, 128.53, 130.76 (2Ph-2-6, C-3a″, C-4″, C-5″, C-6″), 132.16, 133.34 (2Ph-1), 137.02 (5a-C=N), 142.95 (C-7a″), 157.13 (2-C=O), 167.36, 169.56 (8-C=O, COOMe), 175.22 (2″-C=O). HRMS (ESI): *m*/*z* [*M* + H]^+^ calcd for C_34_H_33_N_7_O_5_S: 652.2337; found: 652.2314.

**Ethyl *rac*-(2′*R*,3a*S*,4′*R*,7*S*,9a*R*)-1,1′,3-trimethyl-2,2″,8-trioxo-3a,9a-diphenyl-1,2,3,3a,4,9a-hexahydro-8*H*-dispiro[imidazo[4,5-*e*]thiazolo[2,3-*c*][1,2,4]triazine-7,3′-pyrrolidine-2′,3″-indoline]-4′-carboxylate (11c).** Yield 237 mg (73%); white powder; mp: 253–255 °C. IR (KBr): ν 3341, 3237 (NH), 3067, 3033 (Ar), 2978, 2936, 2875, 2786 (Alk), 1729, 1708, 1652, 1620, 1599 (C=O, C=N) cm^−1^. ^1^H NMR (300 MHz, DMSO-*d*_6_): δ 1.37 (t, *J* = 7.1 Hz, 3H, CH_3_), 2.02 (s, 3H, 1′-NCH_3_), 2.32 (s, 3H, NCH_3_), 2.35 (s, 3H, NCH_3_), 3.29–3.37 (m, 1H, 5′-H), 3.55 (dd, *J* = 9.1, 7.0 Hz, 1H, 5′-H), 4.28–4.37 (m, 2H, OCH_2_), 4.75 (dd, *J* = 10.5, 6.7 Hz, 1H, 4′-H), 6.53–6.56 (m, 2H, Ph-2,6), 6.76–7.15 (m, 11H, Ph-2,6, 2Ph-3-5, 5″-H, 6″-H, 7″-H), 7.63 (d, *J* = 7.6 Hz, 1H, 4″-H), 7.73 (s, 1H, 4-NH), 10.43 (s, 1H, 1″-NH). ^13^C NMR (75 MHz, DMSO-*d*_6_): δ 14.72 (CH_3_), 25.78, 29.60 (2NCH_3_), 35.03 (1′-NCH_3_), 49.53, 51.33 (C-4′, C-5′), 61.88 (OCH_2_), 69.56 (C-7), 78.37, 82.06, 86.41 (C-2′, C-3a, C-9a), 110.50 (C-7″), 121.57, 122.18, 127.38, 127.72, 128.25, 128.40, 128.53, 129.07, 131.41 (2Ph-2-6, C-3a″, C-4″, C-5″, C-6″), 132.56, 133.76 (2Ph-1), 138.40 (5a-C=N), 143.41 (C-7a″), 157.58 (2-C=O), 168.06, 169.63 (8-C=O, COOEt), 175.33 (2″-C=O). HRMS (ESI): *m*/*z* [*M* + H]^+^ calcd for C_34_H_33_N_7_O_5_S: 652.2337; found: 652.2326.

**Ethyl *rac*-(2′*R*,3a*S*,4′*R*,7*S*,9a*R*)-1′-ethyl-1,3-dimethyl-2,2″,8-trioxo-3a,9a-diphenyl-1,2,3,3a,4,9a-hexahydro-8*H*-dispiro[imidazo[4,5-*e*]thiazolo[2,3-*c*][1,2,4]triazine-7,3′-pyrrolidine-2′,3″-indoline]-4′-carboxylate (11d).** Yield 136 mg (41%); white powder; mp: 244–246 °C. IR (KBr): ν 3242, 3141 (NH), 3089, 3035 (Ar), 2997, 2974, 2934, 2866, 2834 (Alk), 1748, 1733, 1714, 1654, 1622, 1600 (C=O, C=N) cm^−1^. ^1^H NMR (300 MHz, DMSO-*d*_6_): δ 0.94 (t, *J* = 7.2 Hz, 3H, CH_3_), 1.38 (t, *J* = 7.1 Hz, 3H, CH_3_), 2.11–2.18 (m, 1H, 1′-NCH_2_), 2.24–2.30 (m, 1H, 1′-NCH_2_), 2.33 (s, 3H, NCH_3_), 2.37 (s, 3H, NCH_3_), 3.25–3.32 (m, 1H, 5′-H), 3.59 (t, *J* = 8.2 Hz, 1H, 5′-H), 4.29–4.39 (m, 2H, OCH_2_), 4.78 (dd, *J* = 10.7, 7.0 Hz, 1H, 4′-H), 6.54–6.57 (m, 2H, Ph-2,6), 6.76 (d, *J* = 7.6 Hz, 1H, 7″-H), 6.97–7.34 (m, 10H, Ph-2,6, 2Ph-3-5, 5″-H, 6″-H), 7.64 (d, *J* = 6.5 Hz, 1H, 4″-H), 7.73 (s, 1H, 4-NH), 10.41 (s, 1H, 1″-NH). ^13^C NMR (75 MHz, DMSO-*d*_6_): δ 13.49, 14.21 (2CH_3_), 25.27, 29.10 (2NCH_3_), 42.66 (1′-NCH_2_), 48.65 (C-4′, C-5′), 61.36 (OCH_2_), 68.88 (C-7), 77.71, 81.53, 86.93 (C-2′, C-3a, C-9a), 109.95 (C-7″), 121.01, 122.18, 126.89, 127.20, 127.73, 127.78, 128.00, 128.55, 130.82 (2Ph-2-6, C-3a″, C-4″, C-5″, C-6″), 132.10, 133.34 (2Ph-1), 137.76 (5a-C=N), 143.06 (C-7a″), 157.05 (2-C=O), 167.49, 169.21 (8-C=O, COOEt), 175.20 (2″-C=O). HRMS (ESI): *m*/*z* [*M* + H]^+^ calcd for C_35_H_35_N_7_O_5_S: 666.2493; found: 666.2488.

**Ethyl *rac*-(2′*R*,3a*S*,4′*R*,7*S*,9a*R*)-5″-bromo-1,1′,3-trimethyl-2,2″,8-trioxo-3a,9a-diphenyl-1,2,3,3a,4,9a-hexahydro-8*H*-dispiro[imidazo[4,5-*e*]thiazolo[2,3-*c*][1,2,4]triazine-7,3′-pyrrolidine-2′,3″-indoline]-4′-carboxylate (11e).** Yield 171 mg (47%); white powder; mp: 231–232 °C. IR (KBr): ν 3342, 3270 (NH), 3058 (Ar), 2948, 2876, 2791 (Alk), 1732, 1658, 1618 (C=O, C=N) cm^−1^. ^1^H NMR (300 MHz, DMSO-*d*_6_): δ 1.35 (t, *J* = 7.1 Hz, 3H, CH_3_), 2.06 (s, 3H, 1′-NCH_3_), 2.42 (s, 3H, NCH_3_), 2.46 (s, 3H, NCH_3_), 3.29–3.34 (m, 1H, 5′-H), 3.55 (t, *J* = 8.3 Hz, 1H, 5′-H), 4.26–4.39 (m, 2H, OCH_2_), 4.76 (dd, *J* = 10.7, 7.0 Hz, 1H, 4′-H), 6.49–6.59 (m, 2H, Ph-2,6), 6.77 (d, *J* = 8.3 Hz, 1H, 7″-H), 6.81–7.49 (m, 8H, 2Ph-3-5, Ph-2,6), 7.52 (dd, *J* = 8.3, 2.1 Hz, 1H, 6″-H), 7.78–7.79 (m, 2H, 4″-H, 4-NH), 10.67 (s, 1H, 1″-NH). ^13^C NMR (75 MHz, DMSO-*d*_6_): δ 14.20 (CH_3_), 25.11, 29.64 (2NCH_3_), 34.64 (1′-NCH_3_), 48.36, 50.83 (C-4′, C-5′), 61.30 (OCH_2_), 68.54 (C-7), 78.04, 81.29, 86.75 (C-2′, C-3a, C-9a), 112.19 (C-7″), 113.01, 123.81, 127.01, 127.23, 127.74, 128.20, 128.64, 129.99, 131.73 (2Ph-2-6, C-3a″, C-4″, C-5″, C-6″), 133.25, 133.91 (2Ph-1), 136.58 (5a-C=N), 142.18 (C-7a″), 156.97 (2-C=O), 167.39, 168.98 (8-C=O, COOEt), 174.24 (2″-C=O). HRMS (ESI): *m*/*z* [*M* + H]^+^ calcd for C_34_H_32_BrN_7_O_5_S: 730.1442; found: 730.1434.

**Ethyl *rac*-(2′*R*,3a*S*,4′*R*,7*S*,9a*R*)-6″-chloro-1,1′,3-trimethyl-2,2″,8-trioxo-3a,9a-diphenyl-1,2,3,3a,4,9a-hexahydro-8*H*-dispiro[imidazo[4,5-*e*]thiazolo[2,3-*c*][1,2,4]triazine-7,3′-pyrrolidine-2′,3″-indoline]-4′-carboxylate (11f).** Yield 239 mg (70%); white powder; mp: 239–241 °C. IR (KBr): ν 3326, 3268 (NH), 3062, 3041 (Ar), 2968, 2940, 2873, 2850, 2794 (Alk), 1732, 1675, 1612 (C=O, C=N) cm^−1^. ^1^H NMR (300 MHz, DMSO-*d*_6_): δ 1.36 (t, *J* = 7.1 Hz, 3H, CH_3_), 2.05 (s, 3H, 1′-NCH_3_), 2.38 (s, 3H, NCH_3_), 2.48 (s, 3H, NCH_3_), 3.28–3.34 (m, 1H, 5′-H), 3.57 (t, *J* = 8.1 Hz, 1H, 5′-H), 4.28–4.37 (m, 2H, OCH_2_), 4.73 (dd, *J* = 10.7, 6.8 Hz, 1H, 4′-H), 6.45–6.65 (m, 2H, Ph-2,6), 6.80 (s, 1H, 7″-H), 6.85–7.45 (m, 9H, 2Ph-3-5, Ph-2,6, 5″-H), 7.63 (d, *J* = 8.2 Hz, 1H, 4″-H), 7.77 (s, 1H, 4-NH), 10.63 (s, 1H, 1″-NH). ^13^C NMR (75 MHz, DMSO-*d*_6_): δ 14.18 (CH_3_), 25.06, 29.60 (2NCH_3_), 34.52 (1′-NCH_3_), 48.42, 50.79 (C-4′, C-5′), 61.29 (OCH_2_), 68.53 (C-7), 77.83, 81.26, 86.64 (C-2′, C-3a, C-9a), 110.62 (C-7″), 120.47, 120.88, 126.96, 127.22, 127.72, 128.16, 128.60, 129.06 (2Ph-2-6, C-3a″, C-4″, C-5″), 131.79, 133.20, 135.61 (2Ph-1, 6″-CCl), 136.68 (5a-C=N), 144.25 (C-7a″), 156.97 (2-C=O), 167.39, 168.98 (8-C=O, COOEt), 174.24 (2″-C=O). HRMS (ESI): *m*/*z* [*M* + H]^+^ calcd for C_34_H_32_ClN_7_O_5_S: 686.1947; found: 686.1941.

## 4. Conclusions

In conclusion, we have developed effective and highly regio- and diastereoselective methods for the synthesis of two series of regioisomeric polyheterocyclic compounds incorporated oxindole and imidazothiazolotriazine fragments spiro-linked with the pyrrolidine cycle via a [3+2] cycloaddition of azomethine ylides to ylidene derivatives of imidazothiazolotriazines. It has been shown that upon treatment with sodium alkoxides, the synthesized compounds has been isomerized into other diastereomers inaccessible with the direct cycloaddition reaction which makes it possible to obtain two more series of diastereoisomeric dispiro[imidazothiazolotriazine-pyrrolidin-oxindoles] from the same starting compounds. The methods allow to prepare a wide series of different diastereomers of target compounds for further investigations. 

## Data Availability

The data presented in this study are available in the article and in the Appendix A.

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
