# Peer review of "Diastereoselective Synthesis of Dispiro[Imidazothiazolotriazine-Pyrrolidin-Oxindoles] and Their Isomerization Pathways in Basic Medium"

_ijms, 2023, doi:10.3390/ijms242216359_

Round 1

Reviewer 1 Report

Comments and Suggestions for Authors

The main topic of the paper, however, the work require important change based on the actual state of nowledge in the area of [3+2] cycloaddition theory. In particular:

Terms such as "dipolar cycloaddition" are completly outdated baceuse many examples of "1,3-dipoles" exhibit not the polar nature, but carbenoid, biradical etc. Term "[3+2] cycloaddition should be consequently used along the manuscript.

Consequently as previous, terme such as "dipolarophile" should be replaced to more appropriate ("ethylenic component" or "2pi-component" for example)

Scheme 2 and 3 suggest the one-step nature of these cycloadditions. Unfortunately, any experiments in the mechanistic area were not performed. So, the scheme should be modified. Next, information about theoretically possible mechanisms (non polar one step, non-polar stepwise, polar one-step, polar one-step two-stage, polar zwitterionic) should be added woth respective examples from the recent literature.

Author Response

The authors kindly thank the reviewers for their comments.

Referee 1.

  1. The main topic of the paper, however, the work require important change based on the actual state of nowledge in the area of [3+2] cycloaddition theory. In particular:
    Terms such as "dipolar cycloaddition" are completly outdated baceuse many examples of "1,3-dipoles" exhibit not the polar nature, but carbenoid, biradical etc. Term "[3+2] cycloaddition should be consequently used along the manuscript.

The term "dipolar cycloaddition" was changed by term "[3+2] cycloaddition” along the manuscript.

  1. Consequently as previous, terme such as "dipolarophile" should be replaced to more appropriate ("ethylenic component" or "2pi-component" for example)

The term "dipolarophile" was replaced to more appropriate "ethylenic component".

  1. Scheme 2 and 3 suggest the one-step nature of these cycloadditions. Unfortunately, any experiments in the mechanistic area were not performed. So, the scheme should be modified. Next, information about theoretically possible mechanisms (non polar one step, non-polar stepwise, polar one-step, polar one-step two-stage, polar zwitterionic) should be added with respective examples from the recent literature.

Scheme 2 was modified. We added the Scheme with formation of azomethine ylide and its possible dipolar, zwitterionic or diradical structures. We also added the Scheme of some theoretically possible mechanisms of cycloaddition (polar zwitterionic, non-polar stepwise, polar one-step, non-polar one step). At Scheme 3 (now it is Scheme 5), we das not represent transition states. Dash lines show only topology of forming bonds.

Reviewer 2 Report

Comments and Suggestions for Authors

Authors present a manuscript describing a multicomponent reaction forming complicated heterocyclic strucures. The manuscript might be accepted after having answered all the questions and comments attached in the corresponding pdf.

Author Response

The authors kindly thank the reviewers for their comments.

Referee 2.

  1. At Scheme 3, steric clashes must be indicated.

We made correction in Scheme 3.

  1. At Figure 2 too work-result. Should be schematic, and this one added to the SI.

Figure 2 was moved to the SI.

  1. To Scheme 6: computations, energetics?

We didn,t carry out computation calculations. After your comment, we started the geometry optimization and calculation of the total energy of diastereoisomers 4, 5 and 6. However, no calculations has been completed to date. We plane to carry out corresponding calculations in the future.

  1. To Figure 3: There are also other signals changing in the spectra and other Hs getting closer or more far from the carbonyl. Why was this H chosen?

This H was chosen because its signals are not overlapped with each other and other signals. At the same time, chemical shift of this proton is due to configuration of C(4’) and  closeness of carbonyl groups of oxindole and thiazolidinone cycles. Indeed, other Hs can serve as indicator.

  1. For the synthesis of diastereomers 4l and 4m: Why is this sequential method? Would it be working also in one step?

Since the particles formed from isatin and amino acid have low stability and short lifespan and corresponding amino acid is sterically hindered, the more longer reaction time is needed. The addition of isatin and amino acid in one step lead to decrease of the conversion of starting compounds and the product yield in this case.

  1. Conclusions: If there is no bioassay in the paper, I suggest to remove this sentence or rephrase more carefully.

The sentence “The methods allow to prepare a wide series of different diastereomers of target compounds for biological activity assay.” was rephrase as follow: “The methods allow to prepare a wide series of different diastereomers of target compounds for further investigations.”

Round 2

Reviewer 1 Report

Comments and Suggestions for Authors

Authors considered my remarks and improved the manuscript accordingly.